# Auto-λ: Disentangling Dynamic Task Relationships

**Shikun Liu**                                                                *shikun.liu17@imperial.ac.uk*
*Dyson Robotics Lab, Imperial College London*

**Stephen James**                                                                    *stepjam@berkeley.edu*
*University of California, Berkeley*

**Andrew J. Davison**                                                           *a.davison@imperial.ac.uk*
*Dyson Robotics Lab, Imperial College London*

**Edward Johns**                                                                  *e.johns@imperial.ac.uk*
*Robot Learning Lab, Imperial College London*

**Reviewed on OpenReview:** *https://openreview.net/forum?id=KKeCMim5VN*

## Abstract

Understanding the structure of multiple related tasks allows for multi-task learning to improve the generalisation ability of one or all of them. However, it usually requires training each pairwise combination of tasks together in order to capture task relationships, at an extremely high computational cost. In this work, we learn task relationships via an automated weighting framework, named Auto-λ. Unlike previous methods where task relationships are assumed to be fixed, i.e., task should either be trained together or not trained together, Auto-λ explores continuous, dynamic task relationships via task-specific weightings, and can optimise any choice of combination of tasks through the formulation of a meta-loss; where the validation loss automatically influences task weightings throughout training. We apply the proposed framework to both multi-task and auxiliary learning problems in computer vision and robotics, and show that Auto-λ achieves state-of-the-art performance, even when compared to optimisation strategies designed specifically for each problem and data domain. Finally, we observe that Auto-λ can discover interesting learning behaviors, leading to new insights in multi-task learning. Code is available at https://github.com/lorenmt/auto-lambda.

## 1  Introduction

Multi-task learning can improve model accuracy, memory efficiency, and inference speed, when compared to training tasks individually. However, it often requires careful selection of training tasks, to avoid *negative transfer*, where irrelevant tasks produce conflicting gradients and complicate the optimisation landscape. As such, without prior knowledge of the underlying relationships between the tasks, multi-task learning can sometimes have worse prediction performance than single-task learning.

We define the *relationship* between two tasks to mean *to what extent these two tasks should be trained together*, following a similar definition in (Zamir et al., 2018; Standley et al., 2020; Fifty et al., 2021). For example, we say that task $A$ is more related to task $B$ than task $C$, if the performance of task $A$ is higher when training tasks $A$ and $B$ together, compared to when training tasks $A$ and $C$ together.

To determine which tasks should be trained together, we could exhaustively search over all possible task groupings, where tasks in a group are equally weighted but all other tasks are ignored. However, this requires training $2^{|\mathcal{T}|} - 1$ multi-task networks for a set of tasks $\mathcal{T}$, and the computational cost for this search can be intractable when $|\mathcal{T}|$ is large. Prior works have developed efficient task grouping frameworks based on heuristics to speed up training, such as using an early stopping approximation (Standley et al., 2020) and

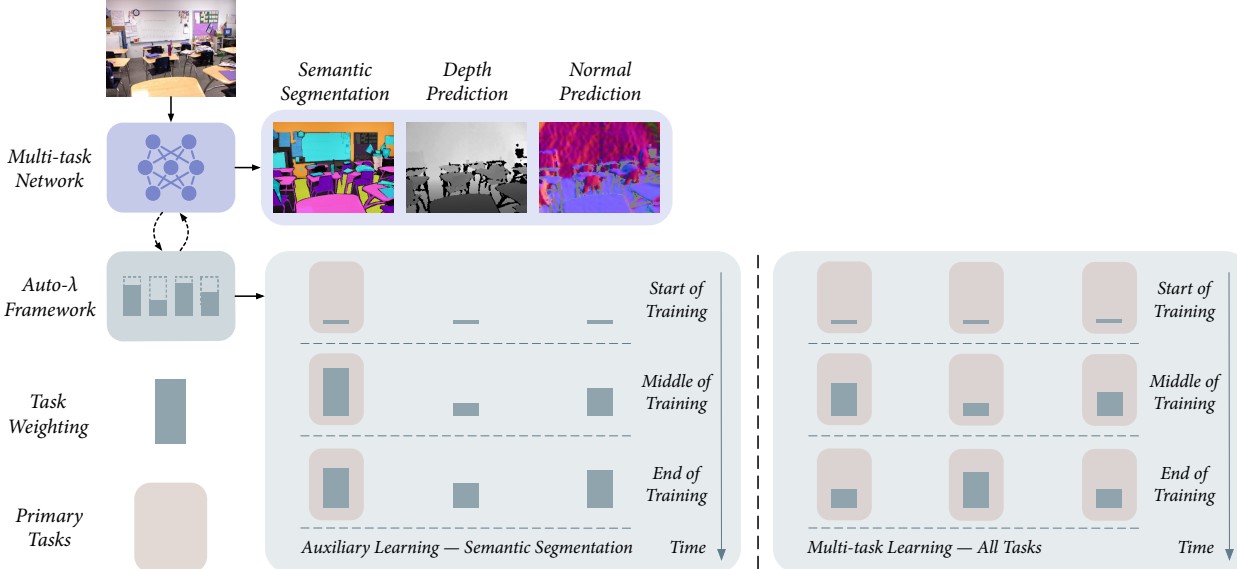

Figure 1: In Auto-$\lambda$, task weightings are dynamically changed along with the multi-task network parameters, in joint optimisation. The task weightings can be updated in both the auxiliary learning setting (one task is the primary task) and the multi-task learning setting (all tasks are the primary tasks). In this example, in the auxiliary learning setting, semantic segmentation is the primary task which we are optimising for. During training, task weightings provide interpretable dynamic task relationships, where high weightings emerge when tasks are strongly related (e.g. normal prediction to segmentation) and low weightings when tasks are weakly related (e.g. depth prediction to segmentation).

computing a lookahead loss averaged across a few training steps (Fifty et al., 2021). However, these task grouping strategies are bounded by two prominent limitations. Firstly, they are designed to be two-stage methods, requiring a *search* for the best task structure and then *re-training* of the multi-task network with the best task structure. Secondly, higher-order task relationships for three or more tasks are not directly obtainable due to high computational cost. Instead, higher-order relationships are approximated by small combinations of lower-order relationships, and thus, as the number of training tasks increases, even evaluating these combinations may become prohibitively costly.

In this paper, instead of requiring these expensive searches or approximations, we propose that the relationship between tasks is *dynamic*, and based on the *current state* of the multi-task network during training. We consider that task relationships could be inferred within a *single* optimisation problem, which runs recurrently throughout training, and automatically balances the contributions of all tasks depending on which tasks we are optimising for. In this way, we aim to *unify* multi-task and auxiliary learning into a single framework — whilst multi-task learning aims to achieve optimal performance for *all* training tasks, auxiliary learning aims to achieve optimal performance for only *a subset of* training tasks (usually only one), which we call the primary tasks, and the rest of the training tasks are included purely to assist the primary tasks.

To this end, we propose a simple meta-learning algorithm, named Auto-$\lambda$. Auto-$\lambda$ explores dynamic task relationships parameterised by task-specific weightings, termed $\lambda$. Through a meta-loss formulation, we use the validation loss of the primary tasks to dictate how the task weightings should be altered, such that the performance of these primary tasks can be improved in the next iteration. This optimisation strategy allows us to jointly update the multi-task network as well as task weightings in a fully end-to-end manner.

We extensively evaluate Auto-$\lambda$ in both multi-task learning and auxiliary learning settings within both computer vision and robotics domains. We show that Auto-$\lambda$ outperforms not only all multi-task and auxiliary learning optimisation strategies, but also the optimal (but static) task groupings we found in the selected datasets. Finally, we take a deep introspection into Auto-$\lambda$'s learning behaviour, and we find that the dynamic relationship between tasks is consistent across numerous multi-task architecture designs, with the converged final relationships aligned with the fixed relationships we found via brute-force search. The simple

and efficient nature of our method leads to a promising new insight towards understanding the structure of tasks, task relationships, and multi-task learning in general.

## 2 Related Work

**Multi-task Architectures**  Multi-Task Learning (MTL) aims at simultaneously solving multiple learning problems while sharing information across tasks. The techniques used in multi-task architecture design can be categorised into hard-parameter sharing (Kokkinos, 2017; Heuer et al., 2021), soft-parameter sharing (Misra et al., 2016; Xu et al., 2018; Liu et al., 2019c; Maninis et al., 2019; Vandenhende et al., 2020), and neural architecture search (Rosenbaum et al., 2018; Gao et al., 2020; Sun et al., 2020).

**Multi-task and Auxiliary-task Optimisation**  In an orthogonal direction to advance architecture design, significant efforts have also been invested to improve multi-task optimisation strategies. Although this is a multi-objective optimisation problem (Sener & Koltun, 2018; Lin et al., 2019; Ye et al., 2021), a single surrogate loss consisting of linear combination of task losses are more commonly studied in practice. Notable works have investigated finding suitable task weightings based on different criteria, such as task uncertainty (Kendall et al., 2018), task prioritisation (Guo et al., 2018) and task loss magnitudes (Liu et al., 2019c). Other works have focused on directly modify task gradients (Chen et al., 2018; 2020; Yu et al., 2020; Javaloy & Valera, 2022; Liu et al., 2021a; Navon et al., 2022).

Similar to multi-task learning, there is a challenge in choosing appropriate tasks to act as auxiliaries for the primary tasks. Du et al. (2018) proposed to use cosine similarity as an adaptive task weighting to determine when a defined auxiliary task is useful. Navon et al. (2021) applied neural networks to optimally combine auxiliary losses in a non-linear manner.

Auto-$\lambda$ is a weighting-based optimisation framework by parameterising these task relationships via learned task weightings. Though these multi-task and auxiliary learning optimisation strategies are encoded to each problem, Auto-$\lambda$ is designed to solve multi-task learning and auxiliary learning in a unified framework.

**Understanding Task Grouping and Relationships**  Prior optimisation methods typically assume all training tasks are somewhat related, and the problem of which tasks should be trained together is often overlooked. In general, task relationships are often empirically measured by human intuition rather than prescient knowledge of the underlying structures learned by a neural network. This motivated the study of task relationships in the transfer learning setting (Zamir et al., 2018; Dwivedi & Roig, 2019). However, Standley et al. (2020) showed that transfer learning algorithms do not carry over to the multi-task learning domain and instead propose a multi-task specific framework to approximate exhaustive search performance. Further work improved the training efficiency for which the task groupings are computed with only a single training run (Fifty et al., 2021). Rather than exploring fixed relationships, our method instead explores dynamic relationships directly during training.

**Meta Learning for Multi-task Learning**  Meta learning (Vilalta & Drissi, 2002; Hospedales et al., 2020) has been often used in the multi-task learning setting, such to generate auxiliary tasks in a self-supervised manner (Liu et al., 2019b; Navon et al., 2021) and improve training efficiency on unseen tasks (Finn et al., 2017; Wang et al., 2021). Our work is also closely related to Kaddour et al. (2020); Liu et al. (2020) which proposed a task scheduler to learn a task-agnostic representation similar to supervised pre-training, whilst ours learns a representation that can adapt specifically to the primary task; Ye et al. (2021) which applied meta learning to solve multi-objective problems, whilst ours focuses on single-objective problems; Michel et al. (2021) which applied meta learning to balance worst-performing tasks, whilst ours balances multi-task learning by finding optimal task relationships. Related to meta learning, our framework is learning to generate suitable and *unbounded* task weightings as a *lookahead* method, optimised based on the validation loss of the primary tasks, as a form of gradient-based meta learning.

**Meta Learning for Hyper-parameter Optimisation**  Since Auto-$\lambda$'s design models multi-task learning optimisation as learning task weightings $\lambda$ dynamically via gradients, we may also consider Auto-$\lambda$ as a meta learning-based hyper-parameter optimisation framework (Maclaurin et al., 2015; Franceschi et al., 2018; Baik

et al., 2020) by treating $\lambda$ as hyper-parameters. Similar to these frameworks, we also formulate a bi-level optimisation problem. However, different to these frameworks, we offer training strategies specifically tailored to the problem of multi-task learning whose goal is not only to obtain good primary task performance, but also explore interesting learning behaviours of Auto-$\lambda$ from the perspective of task relationships.

## 3    Background

**Notations**    We denote a multi-task network to be $f(\cdot;\boldsymbol{\theta})$, with network parameters $\boldsymbol{\theta}$, consisting of task-shared and $K$ task-specific parameters: $\boldsymbol{\theta} = \{\theta_{sh}, \theta_{1:K}\}$. Each task is assigned with task-specific weighting $\boldsymbol{\lambda} = \{\lambda_{1:K}\}$. We represent a set of task spaces by a pair of task-specific inputs and outputs: $\mathcal{T} = \{T_{1:K}\}$, where $T_i = (X_i, Y_i)$.

The design of the task spaces can be further divided into two different settings: a single-domain setting (where all inputs are the same $X_i = X_j, i \neq j$, *i.e.,* one-to-many mapping), and a multi-domain setting (where all inputs are different: $X_i \neq X_j, i \neq j$, *i.e.,* many-to-many mapping). We want to optimise $\boldsymbol{\theta}$ for all tasks $\mathcal{T}$ and obtain a good performance in some pre-selected primary tasks $\mathcal{T}^{pri} \subseteq \mathcal{T}$. If $\mathcal{T}^{pri} = \mathcal{T}$, we are in the multi-task learning setting, otherwise we are in the auxiliary learning setting.

**The Design of Optimisation Methods**    Multi-task or auxiliary learning optimisation methods are designed to balance training and avoid negative transfer. These optimisation strategies can further be categorised into two main directions:

(i) Single Objective Optimisation:

$$\min_{\boldsymbol{\theta}} \sum_{i=1}^{K} \lambda_i \cdot L_i \left( f\left(x_i; \theta_{sh}, \theta_i\right), y_i \right), \tag{1}$$

where the task-specific weightings $\boldsymbol{\lambda}$ are applied for a linearly combined single valued loss. Each task's influence on the network parameters can be *indirectly* balanced by finding a suitable set of weightings which can be manually chosen, or learned through a heuristic (Kendall et al., 2018; Liu et al., 2019c) — which we called *weighting*-based methods; or *directly* balanced by operating on task-specific gradients (Du et al., 2018; Yu et al., 2020; Chen et al., 2018; 2020; Javaloy & Valera, 2022; Liu et al., 2021a; Navon et al., 2022) — which we called *gradient*-based methods. These methods are designed exclusively to alter optimisation.

On the other hand, we also have another class of approaches that determine task groupings (Standley et al., 2020; Fifty et al., 2021), which can be considered as an alternate form of weighting-based method, by finding *fixed and binary* task weightings indicating which tasks should be trained together. Mixing the best of both worlds, Auto-$\lambda$ is an optimisation framework, simultaneously exploring dynamic task relationships.

(ii) Multi-Objective Optimisation:

$$\min_{\boldsymbol{\theta}} \left[ L_i \left( f\left(x_i; \theta_{sh}, \theta_i\right), y_i \right)_{i=1:K} \right]^{\mathsf{T}}, \tag{2}$$

a vector-valued loss which is optimised by achieving Pareto optimality — when no common gradient updates can be found such that all task-specific losses can be decreased (Sener & Koltun, 2018; Lin et al., 2019). Note that, this optimisation strategy can only be used in a multi-task learning setup.

## 4    Auto-$\lambda$: Exploring Dynamic Task Relationships

We now introduce our simple but powerful optimisation framework called Auto-$\lambda$, which explores dynamic task relationships through task-specific weightings.

**The Design Philosophy**    Auto-$\lambda$ is a gradient-based meta learning framework, a unified optimisation strategy for both multi-task and auxiliary learning problems, which learns task weightings, based on *any combination* of primary tasks. The design of Auto-$\lambda$ borrows the concept of *lookahead* methods in meta

learning literature (Finn et al., 2017; Nichol et al., 2018), to update parameters at the current state of learning, based on the observed effect of those parameters on a future state. A recently proposed task grouping method (Fifty et al., 2021) also applied a similar concept, to compute the relationships based on how gradient updates of one task can affect the performance of other tasks, additionally offering the option to couple with other gradient-based optimisation methods. Auto-$\lambda$ however is a standalone framework and encodes task relationships *explicitly* with a set of task weightings associated with training loss, directly optimised based on the validation loss of the primary tasks.

**Bi-level Optimisation** Let us denote $\mathcal{P}$ as the set of indices for all primary tasks defined in $\mathcal{T}^{pri}$; $(x_i^{val}, y_i^{val})$ and $(x_i^{train}, y_i^{train})$ are sampled from the validation and training sets of the $i^{th}$ task space, respectively. The goal of Auto-$\lambda$ is to find optimal task weightings $\boldsymbol{\lambda}^*$, which minimise the validation loss on the primary tasks, as a way to *measure generalisation*, where the optimal multi-task network parameters $\boldsymbol{\theta}^*$ are obtained by minimising the $\boldsymbol{\lambda}^*$ weighted training loss on all tasks. This implies the following bi-level optimisation problem:

$$
\begin{aligned}
\min_{\boldsymbol{\lambda}} \quad & \sum_{i \in \mathcal{P}} L_i(f(x_i^{val}; \theta_{sh}^*, \theta_i^*), y_i^{val}) \\
\text{s.t.} \quad & \boldsymbol{\theta}^* = \arg\min_{\boldsymbol{\theta}} \sum_{i=1}^{K} \lambda_i \cdot L_i(f(x_i^{train}; \theta_{sh}, \theta_i), y_i^{train}).
\end{aligned}
\tag{3}
$$

**Approximation via Finite Difference** Now, we may rewrite Eq. 3 with a simple approximation scheme by updating $\boldsymbol{\theta}$ and $\boldsymbol{\lambda}$ iteratively with one gradient update each:

$$
\boldsymbol{\theta}' = \boldsymbol{\theta} - \alpha \nabla_{\boldsymbol{\theta}} \sum_{i=1}^{K} \lambda_i \cdot L_i(f(x_i^{train}; \theta_{sh}, \theta_i), y_i^{train}),
\tag{4}
$$

$$
\boldsymbol{\lambda} \leftarrow \boldsymbol{\lambda} - \beta \nabla_{\boldsymbol{\lambda}} \sum_{i \in \mathcal{P}} L_i(f(x_i^{val}; \theta_{sh}', \theta_i'), y_i^{val}),
\tag{5}
$$

$$
\boldsymbol{\theta} \leftarrow \boldsymbol{\theta} - \alpha \nabla_{\boldsymbol{\theta}} \sum_{i=1}^{K} \lambda_i \cdot L_i(f(x_i^{train}; \theta_{sh}, \theta_i), y_i^{train}),
\tag{6}
$$

for which $\alpha, \beta$ are manually defined learning rates.

The above optimisation requires computing second-order gradients which may produce large memory and slow down training speed. Therefore, we apply finite difference approximation to reduce complexity, similar to other gradient-based meta learning methods (Finn et al., 2017; Liu et al., 2019a). For simplicity, let's denote $\mathcal{L}(\boldsymbol{\theta}, \boldsymbol{\lambda}), \mathcal{L}^{pri}(\boldsymbol{\theta}, \boldsymbol{\lambda})$ represent $\boldsymbol{\lambda}$ weighted loss produced by all tasks and primary tasks respectively. The gradient to update $\boldsymbol{\lambda}$ can be approximated by:

$$
\begin{aligned}
\nabla_{\boldsymbol{\lambda}} \mathcal{L}^{pri}(\boldsymbol{\theta}^*, \mathbf{1}) &\approx \nabla_{\boldsymbol{\lambda}} \mathcal{L}^{pri}(\boldsymbol{\theta} - \alpha \nabla_{\boldsymbol{\theta}} \mathcal{L}(\boldsymbol{\theta}, \boldsymbol{\lambda}), \mathbf{1}) = \nabla_{\boldsymbol{\lambda}} \mathcal{L}^{pri}(\boldsymbol{\theta}', \mathbf{1}) - \alpha \nabla_{\boldsymbol{\theta}, \boldsymbol{\lambda}}^2 \mathcal{L}(\boldsymbol{\theta}, \boldsymbol{\lambda}) \nabla_{\boldsymbol{\theta}'} \mathcal{L}^{pri}(\boldsymbol{\theta}', \mathbf{1}) \\
&\approx -\alpha \frac{\nabla_{\boldsymbol{\lambda}} \mathcal{L}(\boldsymbol{\theta}^+, \boldsymbol{\lambda}) - \nabla_{\boldsymbol{\lambda}} \mathcal{L}(\boldsymbol{\theta}^-, \boldsymbol{\lambda})}{2\epsilon},
\end{aligned}
\tag{7}
$$

where $\boldsymbol{\theta}' \leftarrow \boldsymbol{\theta} - \alpha \nabla_{\boldsymbol{\theta}} \mathcal{L}(\boldsymbol{\theta}, \boldsymbol{\lambda})$ denotes the network weights for a one-step forward model, and $\boldsymbol{\theta}^{\pm} = \boldsymbol{\theta} \pm \epsilon \cdot \nabla_{\boldsymbol{\theta}'} \mathcal{L}^{pri}(\boldsymbol{\theta}', \mathbf{1})$, with $\epsilon$ a small constant. $\mathbf{1}$ are constants indicating that all primary tasks are of equal importance, and we may also apply different constants based on prior knowledge.

Note that, $\boldsymbol{\lambda}$ is only applied on the training loss not validation loss, otherwise, we would easily reach trivial solutions $\boldsymbol{\lambda} = \mathbf{0}$. In addition, assuming $\boldsymbol{\theta}' = \boldsymbol{\theta}^*$ is also not applicable, otherwise we have $\nabla_{\boldsymbol{\lambda}} = \mathbf{0}$.

**Swapping Training Data** In practice, instead of splitting training data into training and validation sets as in the standard meta learning setup, we sampled training and validation data to be the different batches in the same training dataset. We found that this simple swapping training data strategy can learn similar weightings comparing to sampling batches in different datasets, making Auto-$\lambda$ a single-stage framework.

**Stochastic Task Sampling**  Eq. 4 requires to compute gradients for all training tasks. This may lead to significant GPU memory consumption particularly when the task-shared parameters are accumulating gradients in a multi-domain setting. To further save memory, we optimise $\boldsymbol{\lambda}$ in multiple steps, and for each step, we only compute gradients for $K' \ll K$ tasks sampled stochastically. This design allows Auto-$\lambda$ to be optimised with a *constant memory* independent of the number of training tasks. In practice, we choose the largest possible $K'$ in each dataset that fits in a GPU to speed up training, and we observed that the performance is consistent from a wide range of different $K'$.

## 5  Experiments

To evaluate the generalisation of Auto-$\lambda$, we experimented on both single and multi-domain computer vision and robotics datasets, in multi-task and auxiliary learning settings, with various choices of multi-task architectures.

**Baselines**  In multi-task experiments, we compared Auto-$\lambda$ with state-of-the-art weighting-based multi-task optimisation methods: i) **Equal**: all task weightings are 1, ii) **Uncertainty**  (Kendall et al., 2018): task weightings are optimised via Homoscedastic uncertainty, and iii) **DWA** (Liu et al., 2019c): task weightings are optimised via the rate of change of training losses. In auxiliary learning experiments, we only compared with **GCS** (Gradient Cosine Similarity) (Du et al., 2018) due to the limited works for this setting. Additional experiments comparing to gradient-based methods are further shown in Additional Analysis (Section 7.2).

**Optimisation Strategies**  By default, we considered each single task as the primary task in the auxiliary learning setting, unless labelled otherwise. In all experiments, Auto-$\lambda$'s task weightings were initialised to 0.1, a small weighting which assumes that all tasks are *equally not related*. The learning rate to update these weightings is hand-selected for each dataset. For fair comparison, the optimisation strategies used in all baselines and our method are the same with respect to each dataset and in each data domain. Detailed hyper-parameters are listed in Appendix A.

### 5.1  Results on Dense Prediction Tasks

First, we evaluated Auto-$\lambda$ with dense prediction tasks in **NYUv2** (Nathan Silberman & Fergus, 2012) and **CityScapes** (Cordts et al., 2016), two standard multi-task datasets in a single-domain setting. In NYUv2, we trained on 3 tasks: 13-class semantic segmentation, depth prediction, and surface normal prediction, with the same experimental setting as in Liu et al. (2019c). In CityScapes, we trained on 3 tasks: 19-class semantic segmentation, disparity (inverse depth) estimation, and a recently proposed 10-class part segmentation (de Geus et al., 2021), with the same experimental setting as in Kendall et al. (2018). In both datasets, we trained on two multi-task architectures: **Split**: the standard multi-task learning architecture with hard parameter sharing, which splits at the last layer for the final prediction for each specific task; **MTAN** (Liu et al., 2019c): a state-of-the-art multi-task architecture based on task specific feature-level attention. Both networks were based on ResNet-50 (He et al., 2016) as the backbone architecture.

**Evaluation Metrics**  We evaluated segmentation, depth and normal via mean intersection over union (mIoU), absolute error (aErr.), and mean angle distances (mDist.), respectively. Following Maninis et al. (2019), we also report the overall relative multi-task performance $\Delta_{\text{MTL}}$ of model $m$ averaged with respect to each single-task baseline $b$:

$$\Delta_{\text{MTL}} = \frac{1}{K} \sum_{i=1}^{K} (-1)^{l_i} (M_{m,i} - M_{b,i})/M_{b,i}, \tag{8}$$

where $l_i = 1$ if lower means better performance for metric $M_i$ of task $i$, and 0 otherwise.

**Noise Prediction as Sanity Check**  In auxiliary learning, we additionally trained with a *noise prediction* task along with the standard three tasks defined in a dataset. The noise prediction task was generated by assigning a random noise map sampled from a Uniform distribution for each training image. This

| **NYUv2** | Method | Sem. Seg. [mIoU ↑] | Depth [aErr. ↓] | Normal [mDist. ↓] | $\Delta_{\mathrm{MTL}}$ ↑ | **CityScapes** | Method | Sem. Seg. [mIoU ↑] | Part Seg. [mIoU ↑] | Disp. [aErr. ↓] | $\Delta_{\mathrm{MTL}}$ ↑ |
|---|---|---|---|---|---|---|---|---|---|---|---|
| Single-Task | - | 43.37 | 52.24 | 22.40 | - | Single-Task | - | 56.20 | 52.74 | 0.84 | - |
| Split Multi-Task | Equal | 44.64 | 43.32 | 24.48 | +3.57% | Split Multi-Task | Equal | 54.03 | 50.18 | 0.79 | −0.92% |
|  | DWA | 45.14 | 43.06 | 24.17 | +4.58% |  | DWA | 54.93 | 50.15 | 0.80 | −0.80% |
|  | Uncertainty | 45.98 | 41.26 | 24.09 | +6.50% |  | Uncertainty | 56.06 | **52.98** | 0.82 | +0.86% |
|  | Auto-$\lambda$ | **47.17** | **40.97** | **23.68** | **+8.21%** |  | Auto-$\lambda$ | **56.08** | 51.88 | **0.76** | **+2.56%** |
| Split Auxiliary-Task | Uncertainty | 45.26 | 42.25 | 24.36 | +4.91% | Split Auxiliary-Task | Uncertainty | 55.72 | 52.62 | 0.83 | +0.04% |
|  | GCS | 45.01 | 42.06 | 24.12 | +5.20% |  | GCS | 55.76 | 52.19 | 0.80 | +0.98% |
|  | Auto-$\lambda$ [3 Tasks] | **48.04** | 40.61 | 23.31 | +9.66% |  | Auto-$\lambda$ [3 Tasks] | 56.42 | 52.42 | 0.78 | +2.31% |
|  | Auto-$\lambda$ [1 Task] | 47.80 | **40.27** | **23.09** | **+10.02%** |  | Auto-$\lambda$ [1 Task] | **57.89** | **53.56** | **0.77** | **+4.30%** |
| MTAN Multi-Task | Equal | 44.62 | 42.64 | 24.29 | +4.27% | MTAN Multi-Task | Equal | 55.05 | 50.74 | 0.78 | +0.43% |
|  | DWA | 45.04 | 42.81 | 24.02 | +4.89% |  | DWA | 54.71 | 51.07 | 0.80 | −0.35% |
|  | Uncertainty | 46.41 | 40.94 | 23.65 | +7.69% |  | Uncertainty | 56.28 | **53.24** | 0.82 | +1.16% |
|  | Auto-$\lambda$ | **47.63** | **40.37** | **23.28** | **+9.54%** |  | Auto-$\lambda$ | **56.57** | 52.67 | **0.75** | **+3.75%** |
| MTAN Auxiliary-Task | Uncertainty | 44.56 | 42.21 | 24.26 | +4.55% | MTAN Auxiliary-Task | Uncertainty | 56.13 | 52.78 | 0.83 | +0.38% |
|  | GCS | 44.28 | 44.07 | 24.03 | +3.49% |  | GCS | 55.47 | 52.75 | **0.76** | +2.75% |
|  | Auto-$\lambda$ [3 Tasks] | 47.35 | 40.10 | 23.41 | +9.30% |  | Auto-$\lambda$ [3 Tasks] | 57.64 | 52.77 | 0.78 | +3.25% |
|  | Auto-$\lambda$ [1 Task] | **47.70** | **39.89** | **22.75** | **+10.69%** |  | Auto-$\lambda$ [1 Task] | **58.39** | **54.00** | 0.78 | **+4.48%** |

Table 1: Performance on NYUv2 and CityScapes datasets with multi-task and auxiliary learning methods in Split and MTAN multi-task architectures. Auxiliary learning is additionally trained with a noise prediction task. Results are averaged over two independent runs, and the best results are highlighted in bold.

task is designed to test the effectiveness of different auxiliary learning methods in the presence of useless gradients. We trained from scratch for a fair comparison among all methods in our experiments, following prior works (Kendall et al., 2018; Liu et al., 2019c; Sun et al., 2020).

**Results** Table 1 showed results for CityScapes and NYUv2 datasets in both Split and MTAN multi-task architectures. Our Auto-$\lambda$ outperformed all baselines in multi-task and auxiliary learning settings across both multi-task networks, and has a particularly prominent effect in auxiliary learning setting where it doubles the relative overall multi-task performance compared to auxiliary learning baselines.

We show results for two auxiliary task settings: optimising for just one task (Auto-$\lambda$ [1 Task]), where the other three tasks (including noise prediction) are purely auxiliary, and optimising for all three tasks (Auto-$\lambda$ [3 Tasks]), where only the noise prediction task is purely auxiliary. Auto-$\lambda$ [3 Tasks] has nearly identical performance to Auto-$\lambda$ in a multi-task learning setting, whereas the best multi-task baseline *Uncertainty* achieved notably worse performance when trained with noise prediction as an auxiliary task. This shows that standard multi-task optimisation is susceptible to negative transfer, whereas Auto-$\lambda$ can avoid negative transfer due to its ability to minimise $\lambda$ for tasks that do not assist with the primary task. We also show that Auto-$\lambda$ [1 Task] can further improve performance relative to Auto-$\lambda$ [3 Tasks], at the cost of task-specific training for each individual task.

## 5.2 Results on Multi-domain Classification Tasks

We now evaluate Auto-$\lambda$ on image classification tasks in a multi-domain setting. We trained on CIFAR-100 (Krizhevsky, 2009) and treated each of the 20 'coarse' classes as one domain, thus creating a dataset with 20 tasks, where each task is a 5-class classification over the dataset's 'fine' classes, following Rosenbaum et al. (2018); Yu et al. (2020). For multi-task and auxiliary learning, we trained all methods on a VGG-16 network (Simonyan & Zisserman, 2015) with standard hard-parameter sharing (Split), where each task has a task-specific prediction layer.

**Results** In Table 2, we show classification accuracy on the 5 most challenging domains which had the lowest single-task performance, along with the average performance across all 20 domains. Multi-task learning in this dataset is particularly demanding, since we optimised with a ×20 smaller parameter space per task compared to single-task learning. We observe that all multi-task baselines achieved similar overall performance to single-task learning, due to limited per-task parameter space. However, Auto-$\lambda$ was still able to improve the overall performance by a non-trivial margin. Similarly, Auto-$\lambda$ can further improve performance in the auxiliary learning setting, with significantly higher per-task performance in challenging domains with around $5 - 7\%$ absolute improvement in test accuracy.

| CIFAR-100 | Method | People | Aquatic Animals | Small Mammals | Trees | Reptiles | Avg. |
|---|---|---|---|---|---|---|---|
| Single-Task | - | 55.37 | 68.65 | 72.79 | 75.37 | 75.84 | 82.19 |
| Multi-Task | Equal | **57.73** | 73.59 | 74.41 | 74.64 | 76.69 | 82.46 |
| | Uncertainty | 54.14 | 70.62 | 74.08 | 74.62 | 75.62 | 82.03 |
| | DWA | 55.25 | 71.54 | 74.12 | **75.68** | 76.26 | 82.26 |
| | Auto-$\lambda$ | 57.57 | **74.00** | **75.05** | 75.15 | **77.55** | **83.92** |
| Auxiliary-Task | GCS | 56.45 | 71.05 | 72.93 | 74.45 | 76.29 | 82.58 |
| | Auto-$\lambda$ | **60.89** | **75.70** | **75.64** | **77.38** | **81.75** | **84.92** |

Table 2: Performance of 20 tasks in CIFAR-100 dataset with multi-task and auxiliary learning methods. We report the performance from 5 domains giving lowest single-task performance along with the averaged performance across all 20 domains. Results are averaged over two independent runs, and the best results are highlighted in bold.

### 5.3 Results on Robot Manipulation Tasks

Finally, to further emphasise the generality of Auto-$\lambda$, we also experimented on visual imitation learning tasks within a multi-domain robotic manipulation setting.

To train and evaluate our method, we selected 10 tasks (visualised in Fig. 2) from the robot learning environment, RLBench (James et al., 2020). Training data was acquired by first collecting 100 demonstrations for each task, and then running keyframe discovery following James & Davison (2021), to split the task into a smaller number of simple stages to create a behavioural cloning dataset. Our network takes RGB and point-cloud inputs from 3 cameras (left shoulder, right shoulder, and wrist camera), and outputs a continuous 6D pose and discrete gripper action. To distinguish among each of the tasks, a learnable task encoding is also fed to the network for multi-task and auxiliary learning. Full training details are given in Appendix B.

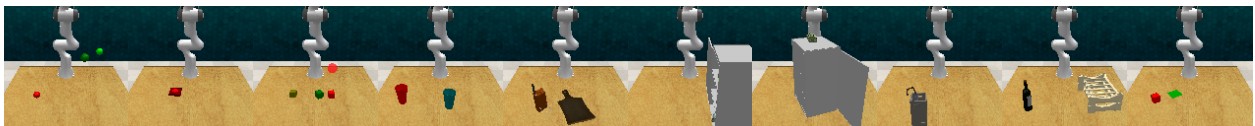

Figure 2: A visual illustration of 10 hand-selected RLBench tasks from the front-facing camera. Task names are: *reach target, push button, pick and lift, pick up cup, put knife on chopping board, take money out of safe, put money in safe, take umbrella out of umbrella stand, stack wine, slide block to target.*

**Results** In Table 3, we reported success rate of each and averaged performance over 10 RLBench tasks. In addition to the baselines outlined in Section 5, we also included an additional baseline based on Priority Replay (Schaul et al., 2016): a popular method for increasing sample efficiency in robot learning systems. For this baseline, prioritisation is applied individually for each task. Similar to computer vision tasks, Auto-$\lambda$ achieved the best performance in both multi-task and auxiliary learning setup, particularly can improved up to $30 - 40\%$ success rate in some multi-stage tasks compared to single-task learning.

| RLBench | Method | Reach Target | Push Button | Pick And Lift | Pick Up Cup | Put Knife on Chopping Board | Take Money Out Safe | Put Money In Safe | Pick Up Umbrella | Stack Wine | Slide Block To Target | Avg. |
|---|---|---|---|---|---|---|---|---|---|---|---|---|
| Single-Task | - | 100 | 95 | 82 | 72 | 36 | 38 | 31 | 37 | 23 | 36 | 55.0 |
| Multi-Task | Equal | **100** | 92 | 86 | 69 | **40** | 57 | 57 | 44 | 16 | 40 | 60.1 |
| | Uncertainty | **100** | 95 | 75 | 56 | 19 | 60 | 79 | 70 | 16 | 65 | 63.5 |
| | DWA | **100** | 90 | **88** | **82** | 35 | **66** | 57 | 61 | 16 | 66 | 66.1 |
| | Priority | **100** | **96** | 78 | 78 | 28 | 52 | 36 | 46 | 15 | 34 | 56.2 |
| | Auto-$\lambda$ | **100** | 95 | 87 | 78 | 31 | 64 | **62** | **80** | 19 | **77** | **69.3** |
| Auxiliary-Task | GCS | **100** | **97** | 81 | 67 | 42 | 56 | 58 | 60 | 14 | 77 | 65.2 |
| | Auto-$\lambda$ | **100** | 93 | **90** | **85** | **49** | **64** | **75** | 74 | **20** | **78** | **72.8** |

Table 3: Performance of 10 RLBench tasks with multi-task and auxiliary learning methods. We reported the success rate with 100 evaluations for each task averaged across two random seeds. Best results are highlighted in bold.

# 6 Intriguing Learning Strategies in Auto-$\lambda$

In this section, we visualise and analyse the learned weightings from Auto-$\lambda$, and find that Auto-$\lambda$ is able to produce interesting learning strategies with interpretable relationships. Specifically, we focus on using Auto-$\lambda$ to understand the underlying *structure* of tasks, introduced next.

## 6.1 Understanding The Structure of Tasks

**Task relationships are consistent.** Firstly, we observe that the structure of tasks is consistent across the choices of learning algorithms. As shown in Fig. 3, the learned weightings with both the NYUv2 and CityScapes datasets are nearly identical, given the same optimisation strategies, independent of the network architectures. This observation is also supported by the empirical findings in Zamir et al. (2018); Standley et al. (2020) in both task transfer and multi-task learning settings.

**Task relationships are asymmetric.** We also found that the task relationships are asymmetric, i.e. learning task *A* with the knowledge of task *B* is not equivalent to learning task *B* with the knowledge of task *A*. A simple example is shown in Fig. 4 Right, where the semantic segmentation task in CityScapes helps the part segmentation task much more than the part segmentation helps the semantic segmentation. This also follows intuition: the representation required for semantic segmentation is a subset of the representation required for part segmentation. This observation is also consistent with multi-task learning frameworks (Lee et al., 2016; 2018; Zamir et al., 2020; Yeo et al., 2021).

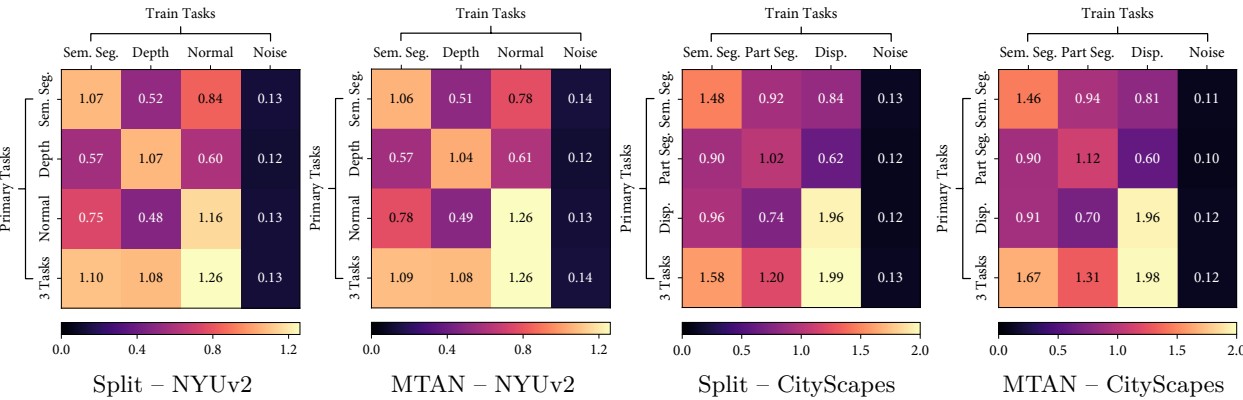

Figure 3: Auto-$\lambda$ explored consistent task relationships in NYUv2 and CityScapes datasets for both Split and MTAN architectures. Higher task weightings indicate stronger relationships, and lower task weightings indicate weaker relationships.

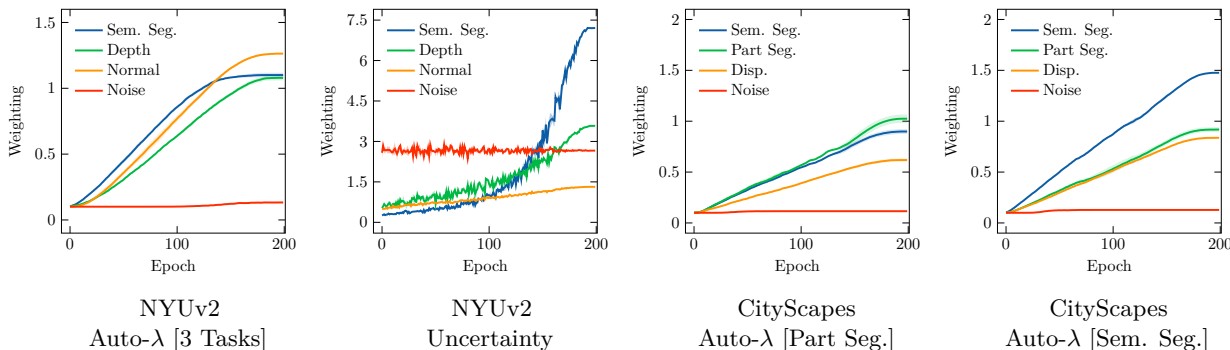

Figure 4: Auto-$\lambda$ learned dynamic relationships based on the choice of primary tasks and can avoid negative transfer. Whilst Uncertainty method is not able to avoid negative transfer, having a constant weighting on noise prediction task across the entire training stage. [·] represents the choice of primary tasks.

**Task relationships are dynamic.** A unique property of Auto-$\lambda$ is the ability to explore dynamic task relationships. As shown in Fig. 4 Left, we can observe a *weighting cross-over* appears in NYUv2 near the end of training, which can be considered as a learning strategy of *automated curricula*. Further, in Fig. 5, we verify that Auto-$\lambda$ achieved higher per-task performance compared to every combination of fixed task groupings in NYUv2 and CityScapes datasets. We can also observe that the task relationships inferred by the fixed task groupings is perfectly aligned with the relationships learned with Auto-$\lambda$. For example, the performance of semantic segmentation trained with normal prediction ($+6.6\%$) is higher than the performance trained with depth prediction ($-6.0\%$), which is consistent with fact that

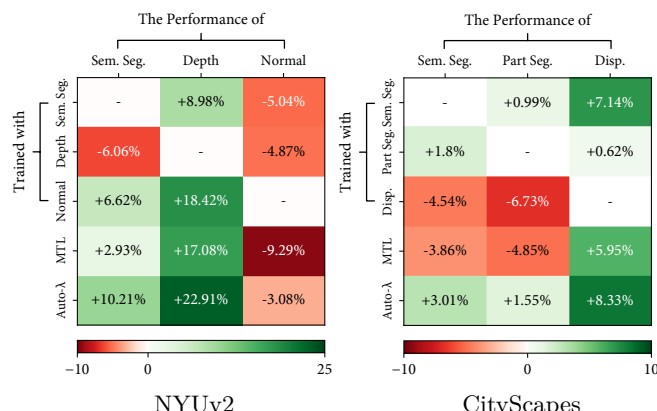

Figure 5: Auto-$\lambda$ achieved best per-task performance compared to every combination of fixed task groupings.

the weighting of normal prediction (0.84) is higher than depth prediction (0.52) as shown in Fig. 3. In addition, we can observe that the Uncertainty method is not able to avoid negative transfer from the noise prediction task, having a constant weighting across the entire training stage, which leads to a degraded multi-task performance as observed in Table 1. These observations confirm that Auto-$\lambda$ is an advanced optimisation strategy, and is able to learn accurate and consistent task relationships.

# 7 Additional Analysis

Finally, we present some additional analyses with Split multi-task architecture to understand the behaviour of Auto-$\lambda$ with respect to different hyper-parameters and other types of optimisation strategies.

## 7.1 Robustness on Training Strategies

Here, we evaluate different hyper-parameters trained with Auto-$\lambda$ [3 Tasks] in the auxiliary learning setting. As seen in Fig. 6, we found that Auto-$\lambda$ optimised with direct second-order gradients offers very similar task weightings compared to when optimised with approximated first-order gradients. In addition, we discovered that using first-order gradients may speed up training time roughly $\times 2.3$. In Table 4, we show that initialising with a small weighting and a suitable learning rate is important to achieve a good performance. A larger learning rate leads to saturated weightings which causes unstable network optimisation, and a larger initialisation would not successfully avoid negative transfer. In addition, optimising network parameters and task weightings with different data is also essential which otherwise would slightly decrease performance.

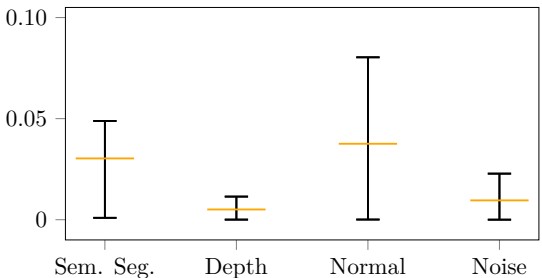

Figure 6: Mean and the range of per-task weighting difference for Auto-$\lambda$ [3 Tasks] optimised with direct and approximated gradients in NYUv2 dataset.

| | Task Weightings | | | | $\Delta_{\mathrm{MTL}}$ |
|---|---|---|---|---|---|
| | Sem. Seg. | Depth | Normal | Noise | |
| $Init = 0.01$ | 0.97 | 0.95 | 1.1 | 0.02 | $+8.98\%$ |
| $Init = 1.0$ | 2.00 | 2.11 | 2.08 | 1.00 | $+1.42\%$ |
| $LR = 3 \cdot 10^{-5}$ | 0.43 | 0.37 | 0.46 | 0.11 | $+8.53\%$ |
| $LR = 3 \cdot 10^{-4}$ | 3.10 | 3.34 | 3.26 | 0.15 | $+8.56\%$ |
| $LR = 1 \cdot 10^{-3}$ | 10.5 | 10.5 | 10.3 | 0.23 | $+5.04\%$ |
| No Swapping | 2.67 | 2.76 | 2.98 | 0.20 | $+8.17\%$ |
| Our Setting | 1.11 | 1.06 | 1.26 | 0.12 | $\mathbf{+9.66\%}$ |

Table 4: Relative multi-task performance in NYUv2 dataset trained with Auto-$\lambda$ [3 Tasks] with different hyper-parameters. The default setting is $Init = 0.1$, $LR = 1 \cdot 10^{-4}$ and with training data swapping.

## 7.2 Comparison to Gradient-based Methods

Since Auto-$\lambda$ is a weighting-based optimisation method, it can naturally be combined with gradient-based methods to further improve performance. We evaluated Auto-$\lambda$ along with the other weighting-based baselines described in Sec. 5, when combined with recently proposed state-of-the-art gradient-based methods designed for multi-task learning: GradDrop (Chen et al., 2020), PCGrad (Yu et al., 2020) and CA-Grad (Liu et al., 2021a). We trained all methods in NYUv2 dataset with standard 3 tasks in the multi-task learning setup.

|           | Equal   | DWA     | Uncertainty | Auto-$\lambda$ |
|-----------|---------|---------|-------------|----------------|
| Vanilla   | +3.57%  | +4.58%  | +6.50%      | **+8.21%**     |
| + GradDrop| +4.65%  | +5.93%  | +6.22%      | **+8.12%**     |
| + PCGrad  | +5.09%  | +4.37%  | +6.20%      | **+8.50%**     |
| + CAGrad  | +7.05%  | +8.08%  | +9.65%      | **+11.07%**    |

Table 5: NYUv2 relative multi-task performance trained with both weighting-based and gradient-based methods in the multi-task learning setting.

In Table 5, we can observe that Auto-$\lambda$ remains the best optimisation method even compared to other gradient-based methods in the vanilla setting (with Equal weighting). Further, combined with a more advanced gradient-based method such as CAGrad, Auto-$\lambda$ can reach even higher performance.

## 7.3 Comparison to Strong Regularisation Methods

Finally, recent works (Lin et al., 2021; Kurin et al., 2022) suggested that many multi-task optimisation methods can be interpreted as forms of implicit regularisation. They showed that when using strong regularisation and stabilisation techniques from single-task learning, training by simply minimising the sum of task losses, or with randomly generated task weightings, can achieve performance competitive with complex multi-task methods.

As such, we now evaluate Auto-$\lambda$, along with all multi-task baselines evaluated in our Experiments section, as well as all multi-task methods included in the original work of (Kurin et al., 2022), coupled with this

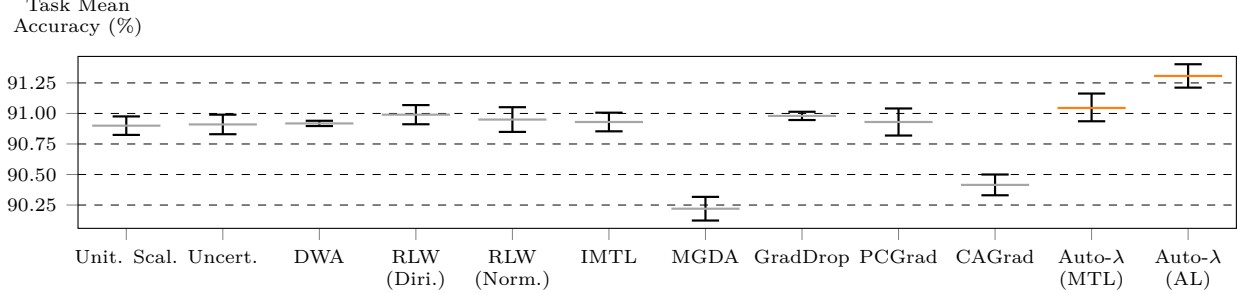

(a) Mean and the range (3 runs) for the averaged task test accuracy

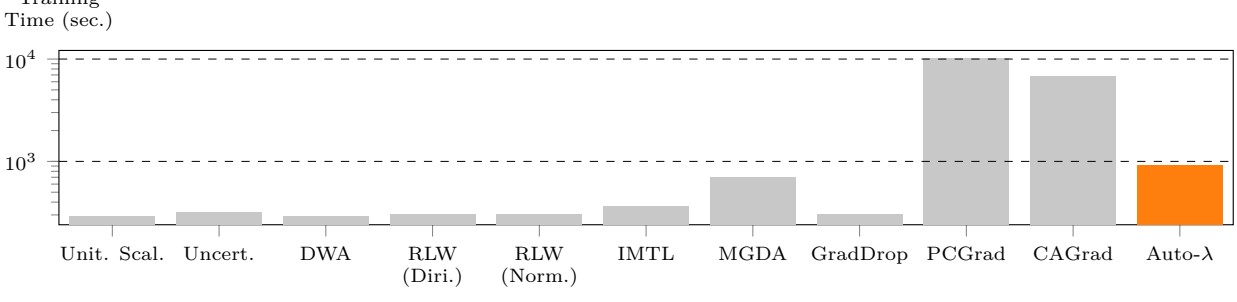

(b) Mean per-epoch training time (10 reptitions)

Figure 7: All multi-task methods perform the same or worse than Unit. Scal. on the CelebA dataset trained with strong regularisation, except Auto-$\lambda$. Part of the results are directly borrow from Kurin et al. (2022).

strong regularisation on CelebA dataset (Liu et al., 2015), for a challenging 40-task classification problem. We trained these multi-task methods with the exact same experimental setting in Kurin et al. (2022) for a fair comparison. To conclude, we compared with: Unit. Scal. (Kurin et al., 2022), DWA (Liu et al., 2019c), RLW (with weights sampled from a Dirichlet and a Normal Distribution) (Lin et al., 2021), IMTL (Liu et al., 2021b), MGDA (Sener & Koltun, 2018), GradDrop (Chen et al., 2020), PCGrad (Yu et al., 2020), CAGrad (Liu et al., 2021a), for a total of 10 multi-task optimisation methods.

To our surprise, though most methods achieve similar performance, which is consistent with the findings in (Kurin et al., 2022), Auto-$\lambda$ is still able to improve performance (marginally in the multi-task learning setting, and significantly in the auxiliary learning setting) with a clear statistical significance. The improvement is especially pronounced in the auxiliary learning mode, which is the unique learning mode of Auto-$\lambda$, showing the multi-task network's generalisation imposed from Auto-$\lambda$ is more than implicit regularisation.

In addition, we also compared training time across these multi-task methods, and we re-scaled the training time in our implementation to Kurin et al. (2022)'s setting for a fair comparison. We can observe that Auto-$\lambda$ requires three times longer the training time than Unit. Scal. (Equal weighting) (Kurin et al., 2022), in consistent with its theoretical design, since Auto-$\lambda$ needs to compute additional two forward and two backward passes to approximate the second-order gradients. Though Auto-$\lambda$ requires longer training time, it can outperform other multi-task methods, and still an order of magnitude faster than some gradient-based methods such as PCGrad (Yu et al., 2020) and CAGrad (Liu et al., 2021a).

## 8  Conclusions, Limitations and Discussion

In this paper, we have presented Auto-$\lambda$, a unified multi-task and auxiliary learning optimisation framework. Auto-$\lambda$ operates by exploring task relationships in the form of task weightings in the loss function, which are allowed to dynamically change throughout the training period. This allows optimal weightings to be determined at any one point during training, and hence, a more optimal period of learning can emerge than if these weightings were fixed throughout training. Auto-$\lambda$ achieves state-of-the-art performance in both computer vision and robotics benchmarks, for both multi-task learning and auxiliary learning, even when compared to optimisation methods that are specifically designed for just one of those two settings.

For transparency, we now discuss some limitations of Auto-$\lambda$ that we have noted during our implementations, and we discuss our thoughts on future directions with this work.

**Hyper-parameter Search**   To achieve optimal performance, Auto-$\lambda$ still requires hyper-parameter search (although the performance is primarily sensitive to only one parameter, the learning rate, making this search relatively simple). Some advanced training techniques, such as incorporating weighting decay or bounded task weightings, might be helpful to find a general set of hyper-parameters which work for all datasets.

**Training Speed**   The design of Auto-$\lambda$ requires computing second-order gradients, which is computationally expensive. To address this, we applied a finite-difference approximation scheme to reduce the complexity, which requires the addition of only two forward passes and two backward passes. However, this may still be slower than alternative optimisation methods.

**Single Task Decomposition**   Auto-$\lambda$ can optimise on any type of task. Therefore, it is natural to consider a compositional design, where we decompose a single task into multiple small sub-tasks, e.g. to decompose a multi-stage manipulation tasks into a sequence of stages. Applying Auto-$\lambda$ on these sub-tasks might enable us to explore interesting learning behaviours to improve single task learning efficiency.

**Open-ended Learning**   Given the dynamic structure of the tasks explored by Auto-$\lambda$, it would be interesting to study whether Auto-$\lambda$ could be incorporated into an open-ended learning system, where tasks are continually added during training. The flexibility of Auto-$\lambda$ to dynamically optimise task relationships may naturally facilitate open-ended learning in this way, without requiring manual selection of hyper-parameters for each new task.

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

## A    Detailed Training Strategies

For dense prediction tasks, we followed the same training setup with MTAN based on the code that was made publicly available by the authors (Liu et al., 2019c). We trained Auto-$\lambda$ with learning rate $10^{-4}$ and $3 \cdot 10^{-5}$ for NYUv2 and CityScapes respectively.

For multi-domain classification tasks, we trained each and all tasks with SGD momentum with 0.1 initial learning rate, 0.9 momentum, and $5 \cdot 10^{-4}$ weight decay. We applied cosine annealing for learning rate decay trained with total 200 epochs. We set batch size 32 and we trained Auto-$\lambda$ with $3 \cdot 10^{-4}$ learning rate.

For robot manipulation tasks, we trained with Adam with a constant learning rate $10^{-3}$ for 8000 iterations. We set batch size 32 and we trained Auto-$\lambda$ with $3 \cdot 10^{-5}$ learning rate.

## B    Detailed Experimental Setting for Robotic Manipulation Tasks

Naively applying behaviour cloning (e.g. mapping observations to joint velocities or end-effector incremental poses) for robot manipulations tasks often requires thousands of demonstrations (James et al., 2017). To circumvent that, we first pre-processed the demonstrations by running keyframe discovery (James & Davison, 2021); a process that iterates over each of the demo trajectories and outputs the transitions where interesting things happen, *e.g.* change in gripper state, or velocities approach zero. The results of the keyframe discovery is a small number of end-effector poses and gripper actions for each of the demonstrations, essentially splitting the task into a set of simple stages. The goal of our behaviour cloning setup is to predict these end-effector poses and gripper actions for new task configurations. Training data was then acquired by first collecting 100 demonstrations for each task, and then running keyframe discovery, to split the task into a smaller number of simple stages to create our behavioural cloning dataset.

We optimised an encoder-decoder network which takes the inputs of RGB and point-clouds captured by three different cameras (left shoulder, right shoulder and wrist camera), and outputs a continuous 6D pose and a discrete gripper action. The 6D pose is composed of a 3-dimensional vector encoding spatial position and a 4-dimensional vector encoding rotation (parameterised by a unit quaternion); the gripper action is represented by a binary scalar indicating gripper open and close. The position and rotation are learned through two separate decoders. The position decoder predicts attention maps based on RGB images, then we apply spatial (soft) argmax (Levine et al., 2016) on the corresponding point cloud to output a 3D spatial position of the attended

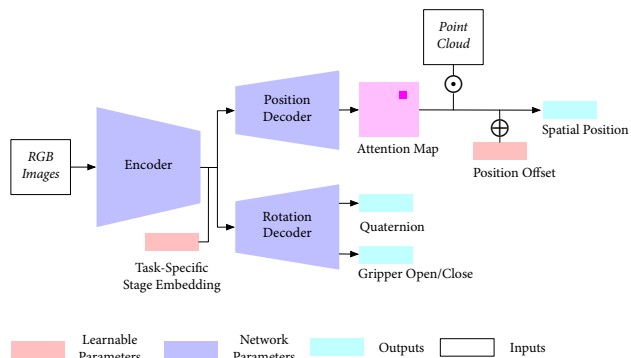

Figure 8: Visualisation of the network design for RL-Bench.

pixel. We additionally optimised a position off-set for each stage of the task, so the predicted position will not be bounded by the position only available in the images. The rotation encoder predicts quaternion and gripper action via direct regression. A learnable task embedding is fed to the network bottleneck for multi-task and auxiliary learning.

## C  Auto-λ Learned Weightings for NYUv2 and CityScapes

We found that the relationships in NYUv2 and CityScapes dataset are usually static from the beginning of training (except for NYUv2 [3 tasks] where we can observe a clear weighting cross-over).

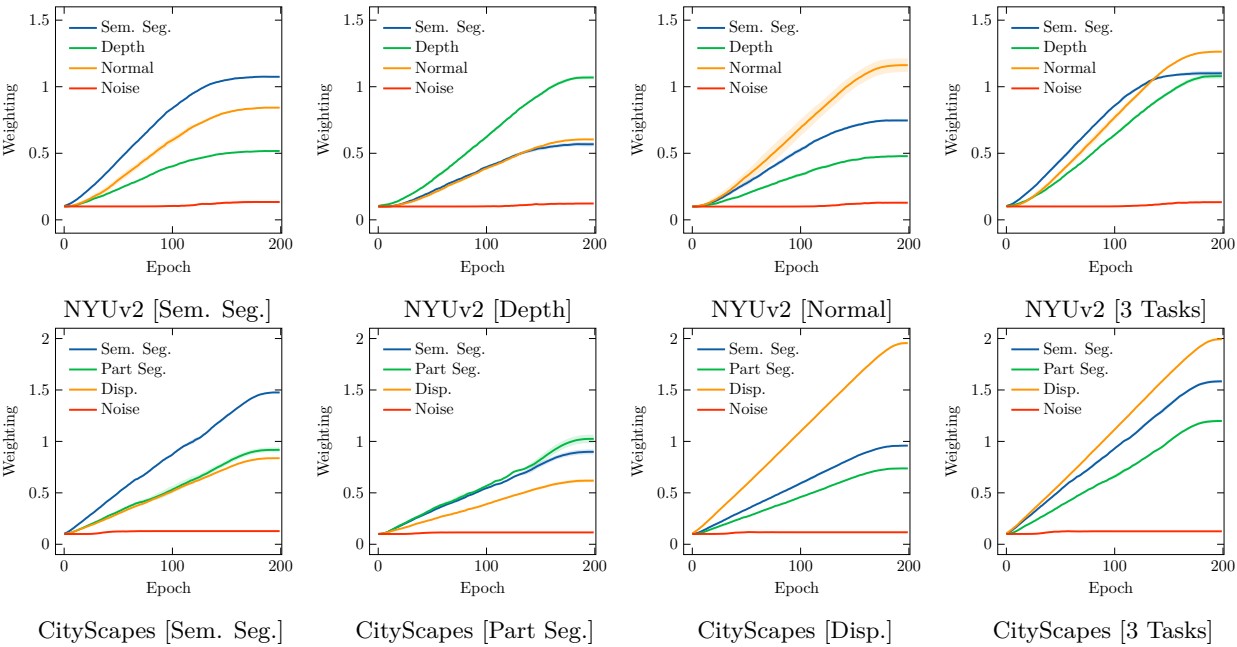

Figure 9: Learning dynamics of Auto-λ optimised on various choices of primary tasks in the auxiliary learning setup with Split architecture.

## D  Auto-λ Learned Weightings for RLBench

The relationships vary more wildly in RLBench tasks, where we can observe multiple weighting cross-over in different training stages.

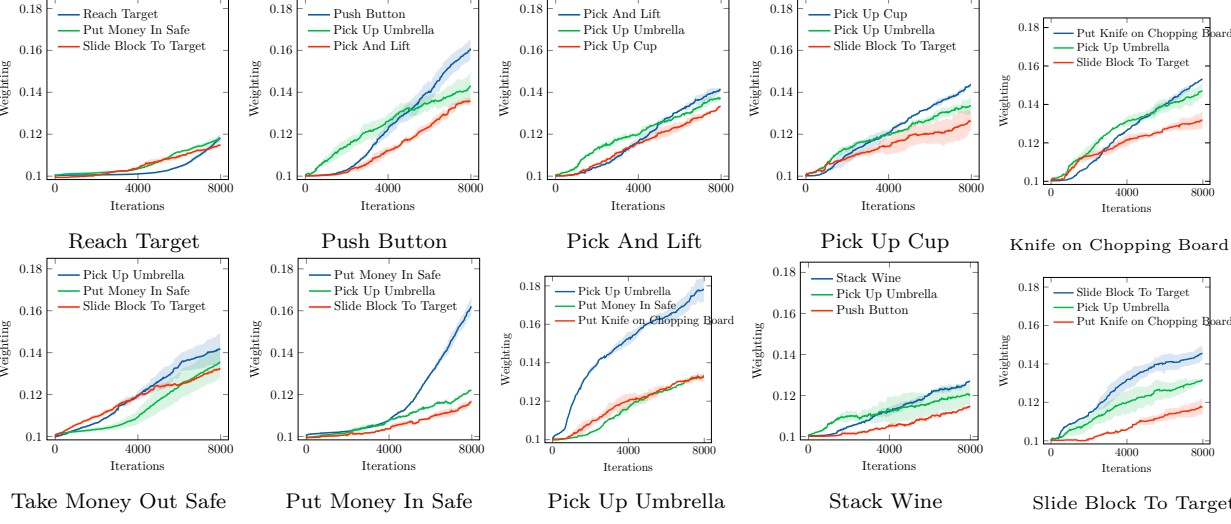

Figure 10: Learning dynamics of Auto-λ optimised on each individual task in the auxiliary learning setup for 10 RLBench tasks. We list 3 tasks with the highest task weightings in each setting.

# E  Auto-$\lambda$ Learned Weightings for CIFAR-100

Interestingly, in the multi-task learning setting of multi-domain classification tasks (last row of Fig. 11), we can see a clear correlation between task weighting and single task learning performance, where the higher weighting is applied for more difficult domain (with low single task learning performance). For example, '*People*' and '*Vehicles 2*', which have the lowest and highest single task learning performance respectively, were assigned with the lowest and the highest task weightings.

| ID 1 | Aquatic Mammals | ID 2 | Fish | ID 3 | Flowers | ID 4 | Food Containers |
|---|---|---|---|---|---|---|---|
| ID 5 | Fruit and Vegetables | ID 6 | Household Electrical Devices | ID 7 | Household furniture | ID 8 | Insects |
| ID 9 | Large Carnivores | ID 10 | Large Man-made Outdoor Things | ID 11 | large natural outdoor scenes | ID 12 | Large Omnivores and Herbivores |
| ID 13 | Medium-sized Mammals | ID 14 | Non-insect Invertebrates | ID 15 | People | ID 16 | Reptiles |
| ID 17 | Small Mammals | ID 18 | Trees | ID 19 | Vehicles 1 | ID 20 | Vehicles 2 |

Table 6: The description of each domain ID in multi-domain CIFAR-100 dataset.

| CIFAR-100 | Method | ID 1 | ID 2 | ID 3 | ID 4 | ID 5 | ID 6 | ID 7 | ID 8 | ID 9 | ID 10 |
|---|---|---|---|---|---|---|---|---|---|---|---|
| Single-Task | - | 68.65 | 81.00 | 82.34 | 83.71 | 89.10 | 88.72 | 84.75 | 85.88 | 87.07 | 90.15 |
| Multi-Task | Equal | 73.59 | 82.36 | 79.78 | 83.94 | 89.14 | 87.03 | 83.73 | 85.87 | 86.67 | 89.86 |
| | Uncertainty | 70.62 | 81.01 | 80.46 | 83.59 | 88.06 | 86.83 | 82.96 | 86.46 | 87.40 | 89.58 |
| | DWA | 71.54 | 82.12 | 81.60 | 83.22 | 89.70 | 86.64 | 82.57 | 86.17 | 87.34 | 90.19 |
| | Auto-$\lambda$ | 74.00 | 83.96 | 81.30 | 83.57 | 88.69 | 87.85 | 84.57 | 87.75 | 88.04 | 92.03 |
| Auxiliary-Task | GCS | 71.05 | 82.27 | 80.31 | 83.36 | 87.07 | 85.94 | 83.05 | 86.80 | 87.54 | 89.34 |
| | Auto-$\lambda$ | 75.70 | 84.39 | 82.71 | 84.64 | 90.23 | 88.02 | 85.52 | 87.36 | 89.04 | 92.20 |
| | Method | ID 11 | ID 12 | ID 13 | ID 14 | ID 15 | ID 16 | ID 17 | ID 18 | ID 19 | ID 20 |
| Single-Task | - | 89.76 | 84.88 | 90.33 | 84.41 | 55.37 | 75.84 | 72.79 | 75.37 | 91.48 | 94.69 |
| Multi-Task | Equal | 89.21 | 86.40 | 89.45 | 85.52 | 57.73 | 76.69 | 74.41 | 74.64 | 90.64 | 94.21 |
| | Uncertainty | 89.80 | 87.07 | 89.76 | 85.64 | 54.14 | 75.62 | 74.08 | 74.62 | 90.83 | 89.54 |
| | DWA | 89.08 | 85.91 | 89.39 | 85.15 | 55.25 | 76.26 | 74.12 | 75.68 | 90.95 | 94.33 |
| | Auto-$\lambda$ | 90.05 | 88.00 | 91.25 | 84.98 | 57.57 | 77.55 | 75.05 | 75.15 | 91.87 | 95.19 |
| Auxiliary-Task | GCS | 89.80 | 85.59 | 89.41 | 85.70 | 56.45 | 76.29 | 72.93 | 74.45 | 90.31 | 93.98 |
| | Auto-$\lambda$ | 90.82 | 87.32 | 90.76 | 86.56 | 60.89 | 81.75 | 75.64 | 77.38 | 91.58 | 95.87 |

Table 7: The complete performance of 20 tasks in multi-domain CIFAR-100 dataset with multi-task and auxiliary learning methods.

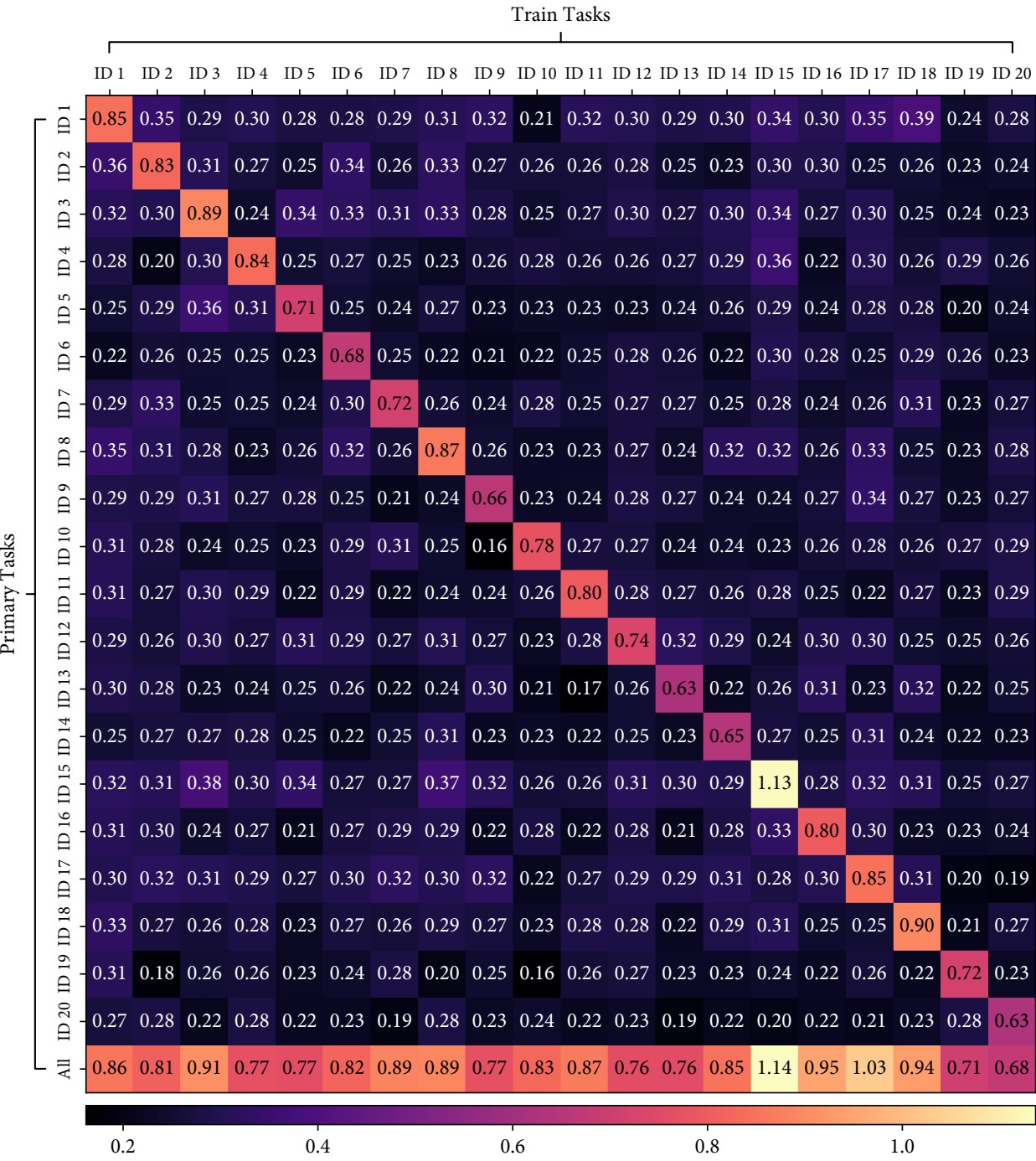

Figure 11: Visualisation of learned weightings in Auto-$\lambda$ in auxiliary learning and multi-task learning setup.

