# OpenReview forum: "Auto-Lambda: Disentangling Dynamic Task Relationships"
_TMLR — Accepted by TMLR_

### Review · Reviewer_EZCk · 2022-04-17

**Summary Of Contributions:**

This paper proposed one method Auto-λ to learn the dynamic task relationships in the multi task learning. It assumes the task relationships should be dynamic w.r.t. the primary tasks, then MTL and auxiliary learning are unified into one framework.  A meta-loss formulation is designed to optimize the parameters in this framework, i.e., multi-task architectures and the task relationships λ. The experimental results demonstrate that both MTL and auxiliary learning problems can benefit from this optimization framework.



**Broader Impact Concerns:**

not applicable.

**Requested Changes:**

Please check Cons and "Detailed Comments" in "Strengths And Weaknesses".

**Strengths And Weaknesses:**

Pros:
This paper proposed to learn dynamic task relationships considering the different primary tasks. Then the MTL and auxiliary learning can be unified., which is interesting and efficient in MTL. Extensive experimental results have been conducted on two domains with different settings. The results demonstrate the effectiveness of the proposed method.

Cons:
1. The position of this paper is not clear.

a). Plain MAML takes equal importance for all tasks involved, which may not be true in real applications. However, there are many works towards learning the task scheduler into meta-learning, e.g., [1,2]. It would be better to provide a comprehensive review on this part.

b). The dynamic task relationship is not well defined and discussed. This paper aims to learn the dynamic task relationships. However, this assumption is not well proved in this paper, either in theoretical or experimental. What are the disadvantages when using the fixed task relationships, and what are the challenges when addressing this problem?

Besides, In the introduction, considering the time consumption in the existing task clustering approaches, the Auto-λ is proposed to learn the dynamic task relationships with the gradient descent. There lacks an efficiency comparison between these methods, i.e., gradient-based methods and the task grouping approaches, and the proposed Auto-λ in the experiments. Moreover, it is unclear why these gradient-based methods cannot be applied to learn the dynamic λ.

2. The technical contribution is limited.

a). All components in this paper are well studied, e.g., single-objective optimization in MTL, gradient-based task relationship design, bi-level optimization, etc. Combining these techniques is straightforward and easy to obtain. The insights of combining them are not clear.

b). The optimization of task relationships λ and the architecture parameters θ are disentangled, and then it is transformed into a bi-level optimization problem, which is trivial and easy to be designed. In section 2, there lacks a comparison between Auto-λ and the existing gradient-based methods mentioned, and the comparisons of the single objective in Eq. (1) and the bi-level optimization mechanism in Eq. (3) are ignored. It is unclear what is the key contribution of the proposed algorithm.

3. Correctness of the algorithm. Existing methods [3,4] point out the efficiency and reliability issues in the bi-level optimization. Therefore, it would be better to provide the theoretical analysis on the correctness of the gradient calculation.


Detailed Comments:
1. As introduced in Section 2, the multi-task architectures can be classified into three classes. While the NAS split is ignored in Tb. 1.
2. It would be better to make a comparison with the existing few-shot learning methods in Section 2.
3. In Table 2, the uncertainty method in the Auxiliary task is missed compared with Tb. 1.
4. In Fig 3, it is not accurate to say that the task relationship is consistent based on the same optimization strategy, since only the proposed optimization algorithm is used in this figure. It would be better to provide the task relationships learned by different methods.
5. The results of Fig. 5 cannot be aligned with the learned relationships in Fig. 3 except for the used example.


Refs
[1] Meta-learning with an Adaptive Task Scheduler.
[2] Probabilistic Active Meta-Learning
[3] A Single-Timescale Method for Stochastic Bilevel Optimization.
[4] A Generic First-Order Algorithmic Framework for Bi-Level Programming Beyond Lower-Level Singleton

---

> ### Author Response · Authors · 2022-04-27
> **Response for R5**
>
> We thank R5 for the constructive suggestions, and we hope to resolve your questions in the following comments.
>
> -- ***Connections to Meta-learning based Task Scheduler***
>
> We agree that our paper is marginally related to the linked papers [1, 2], since we are all learning a task scheduler. However, the problem setting is completely different. In both [1, 2], the task scheduler is learning to sample training tasks during meta-training process, so it can achieve a better generlisation of a meta model on **unseen** tasks. However, in our setting, the task scheduler is learning to re-weight training tasks such that it can have a strong generalisation on primary tasks, which are **known** tasks from the training tasks. And therefore [1, 2] are **not learning task relationships nor alleviate negative transfer** (which are the focus of our work), but to achieve a general representation as a form of supervised pre-training.
>
> -- ***Dynamic Task Relationships***
>
> As similar to the comments we made for R2, we have included a proper definition of task relationships in the updated paper (second paragraph Section 1). Here, the relationship is defined by how the performance of a task can be improved when co-training with another task. If the performance increases, we say these two tasks are strongly related, otherwise they are weakly related. We have an example in our introduction Figure 1. The limitation of fixed task grouping has been explained in the third paragraph of the original Introduction section -- it's computationally expensive and requires re-training the multi-task model after searching the best task grouping based on the choice of primary tasks. We also empirically validate that Auto-Lambda achieves best per-task performance compared to the best combination of fixed task grouping in Fig 5.
>
> -- ***Technical Contribution***
>
> The insights of Auto-Lambda formulation is based on the fact that the learned weightings can be dynamically changed, learning a form of dynamic task relationships. We have confirmed that **our findings are consistent with existing task grouping frameworks in the entire Section 6, though our method is simpler and easier to train**. We agree that each component is well-studied in the literature, but we'd like to highlight our contribution is providing such perspective to **unify multi-task learning and auxiliary into a single optimisation problem** which can be solved with existing tools. This perspective is unique and novel in the MTL community.
>
> In addition, **we have already compared to single objective (weighting-based) optimisation methods (Section 5) as well as gradient-based methods in Section 7.2, all in our original paper.**
>
> -- ***Detailed Comments***
>
> 1. The focus of this work is proposing a new MTL optimisation method, rather than a new MTL architecture. We have evaluated Auto-Lambda with multiple architectures in different datasets to confirm that our method is general and robust to the choice of MTL architectures.
>
> 2. Our method is designed specifically for MTL. It's not suitable for few-shot learning applications.
>
> 3. Uncertainty is **not** an auxiliary learning optimisation method. We included Uncertainty in Table 1, simply showing that **it cannot alleviate negative transfer**, which is a unique property of Auto-Lambda.
>
> 4. To clarify, we did not to intend to suggest that task relationship is consistent based on **the same optimization strategy**. What we say in the paper Section 6.1 first paragraph is that **task relationship is consistent across the choices of learning algorithms.** This can be shown in Fig 3. -- both MTAN and hard parameter sharing learn **near-identical task weightings**.
>
> 5. The results in Fig 5. are aligned with the learned relationships in Fig 3, i.e.. *higher task weightings indicate closer relationships*. For example, The performance of Depth trained with Normal (+18.42) is higher than the performance of Depth trained with Semantic Segmentation (+8.98), aligned with the fact that the weighting of Normal (0.6) is higher than the weighting of Semantic Segmentation (0.57) when having Depth as the primary task. Similarly, the performance of Disp. trained with Semantic Segmentation (+7.14) is higher than the performance of Disp. trained with Part Segmentation (+0.62), aligned with the fact that the weighting of Semantic Segmentation (1.48) is higher than the weighting of Part Segmentation (0.94) when having Disp. as the primary task.
>
> **Update R5**: We have improved the Related Work section: adding discussion with meta-learning based task scheduler **[Update B]**. We have improved paper details: adding definition of task relationships **[Update A]**.

---

### Review · Reviewer_QXX5 · 2022-04-17

**Summary Of Contributions:**

The authors propose Auto-$\lambda$, a method which dynamically adjusts the weights of loss functions in a multi-task learning paradigm via an optimization framework similar to gradient based meta learning. In contrast with prior work which focuses on balancing all tasks equally, Auto-$\lambda$ considers the setting where auxiliary tasks may be used to augment the performance of one or more primary tasks. This change of perspective greatly increases the number of real-world problems Auto-$\lambda$ may be applied to, and in my view, represents an important (and perhaps necessary) shift in the development of future multi-task learning methods. The author's proposed method appears well motivated -- grounded in meta learning, multi-task training augmentation, and multi-task grouping algorithms -- and the empirical analysis on 4 separate datasets is convincing. Finally, the authors offer several ablation experiments (i.e. capacity to handle a "noise" task, asymmetry in task relationships, etc.) in attempting to contribute the community's knowledge of multi-task learning training dynamics and underlying structure.



**Broader Impact Concerns:**

I have no Broader Impact concerns related to this work.

**Requested Changes:**

It seems like there may be an issue with Eq. 7?

Representing the gradient w.r.t. $\lambda$ after first updating the parameters $\theta$, we have (line 1):

$\nabla_{\lambda}\mathcal{L}^{pri}(\theta - \alpha \nabla_{\theta} \mathcal{L}(\theta, \lambda), 1)$

With substitution $\theta' = \theta - \alpha\nabla_{\theta}\mathcal{L}(\theta, \lambda)$, line 2 becomes:

$\nabla_{\lambda}\mathcal{L}^{pri}(\theta - \alpha\nabla_{\theta}\mathcal{L}(\theta, \lambda), 1) - \alpha\nabla^2_{\theta, \lambda}\mathcal{L}(\theta, \lambda)\nabla_{\theta'}\mathcal{L}^{pri}(\theta', 1)$

and then $- \alpha\nabla^2_{\theta, \lambda}\mathcal{L}(\theta, \lambda)\nabla_{\theta'}\mathcal{L}^{pri}(\theta', 1) = 0$?

It also may be helpful to make it explicitly clear that the second argument to $\mathcal{L}^{pri}$ is a "1" representing that the $\lambda$'s are set to a constant value in the outer loss function when you define the notation above Eq. 7.

This is a personal nitpick of mine -- and you can entirely ignore this feedback if you disagree with it -- but under section 5 Experiments, you have "To validate the generalization of ...". This phrasing may sound like you've come to the conclusion that Auto-$\lambda$ works before evaluating the method on experiments. Simply changing "validate" to "evaluate" may change the tone of this opening paragraph to be more inquisitive (i.e. we think this could work, but let's evaluate it on several datasets to find out).

**Strengths And Weaknesses:**

It is my view that this is an extremely strong paper. The method itself is well explained and draws heavily from successes in gradient based meta learning to improve multi-task learning performance. It also represents a possible paradigm shift from "how can we maximize performance on all tasks training together" to incorporate greater flexibility in automatically deciding which tasks would be helpful for the training of one or more primary tasks.

The empirical analysis is robust, and although I am unable to locate their code in the supplementary material, I would like to ask if the authors will commit to open sourcing it following conference reviews. Evaluating their method on 4 separate (and somewhat robust/realistic datasets -- i.e. not just MNIST/CIFAR-10), in addition to their comparison with accepted and popular MTL task weighing methods (i.e. Uncertainty Weights, DWA, GCS, etc.) is highly appropriate.

The ablation experiments were also very compelling. The comparison with a "noise task" highlights their method's capabilities in "tuning out" tasks which are unhelpful in the training of others in the network; a formulation which presents difficulty to other task weighing algorithms. The authors also motivate their intuition that task relationships are dynamic, and while I think many mtl researchers/practitioners would agree with this statement given prior work investigating this topic, Auto-$\lambda$ put a new spin on this concept by explicitly comparing against optimal task-grouping methods in the auxiliary task learning domain.

---

> ### Author Response · Authors · 2022-04-27
> **Response for R4**
>
> We would like to thank R4's high appreciation and support of this work.
>
> Yes, we design Auto-Lambda to be a simple framework and a unified solution to a lot of interesting problems in MTL: exploring task relationships, balancing task-specific learning landscapes, deciding which tasks should be trained together, etc. We really hope this work could benefit the MTL community in general, to better understand MTL's training dynamics, and simultaneously providing useful insights on designing more powerful MTL neural architectures.
>
> Here are some other comments for your questions.
>
> -- ***Code Release***
>
> We have now attached the code, available for download now, in the supplementary material.
>
> -- ***Finite Difference Approximation***
>
> Here $\theta'$ represents a one-step forward model, i.e., the $\theta$ has now been updated by one gradient step and no longer tracks the gradients of the original $\theta$ in $L(\theta,\lambda)$. And then we use $\theta^\pm$ which is based on $\theta'$ to approximate the Hessian (second term). And finally, $\nabla_{\lambda}L^{pri}(\theta', 1)=0$, since it has no $\lambda$ in the loss. We have changed $\theta' =$ into $\theta' \leftarrow$ in the updated paper to avoid confusion.
>
> **Update R4:** We agree that emphasising that 1 are constants would be helpful. As suggested, we have changed "validate" to "evaluate". **[Update A]**

---

### Review · Reviewer_BnS4 · 2022-04-18

**Summary Of Contributions:**

This work proposes auto-\lambda, a simple meta learning based approach, to adaptively adjust the weights for different tasks in both the multi-task and auxiliary learning setting. It formulates the weight adjustment as a bi-level optimization problem, where the upper level is to optimize the weights \lambda with respect to the validation loss, and the lower level is to optimize the model parameters \theta to the training loss. At each update step, the proposed algorithm first calculates the weights for each task by solving the bi-level optimization problem (with a simple one-step gradient update), and then updates the model parameters with the obtained task weights.

In practice, Auto-\lambda simply samples data from a different batch in the same training set (rather than a truly held-out validation set) to calculate the validation loss for weight adjustment. This work uses a simple finite difference approximation to avoid computing the higher-order gradients, and uses stochastic task sampling to save memory for problems with a large number of tasks. Experimental results on CV and robotics applications show auto-\lambda can outperform other well-known methods for both multi-task learning and auxiliary learning problems. The dynamic weights obtained by Auto-\lambda can also be used for analyzing the relationship among tasks in multi-task learning.


**Requested Changes:**

I believe the critical changes will mainly depend on how the authors tackle the **concerns raised above**. Generally, I will expect:
- more discussion/comparison with related works;
- deeper analyses of the proposed Auto-\lambda method (might with additional experiments);

to support the claims made in this work.

In addition to the critical changes, I am also interested to see more discussions or experiments/analyses on:

- The trade-off (performance v.s. training speed) between exact the second-order gradient and the approximate finite difference to solve the bi-level optimization problem at each step;
- The trade-off for stochastic task sampling. Will taking K' << K significantly slow down the training process (in terms of #epochs)? Will the sampling make the training unstable and indeed hurt the final performance (even with the same computation budget)?

Minor Issues:

- The code is not attached although it says so in the introduction. Correct me if I miss it.

- The year of CAGrad (Liu et al 2018) should be 2021?

**Strengths And Weaknesses:**

**Guideline:**

Following the TMLR guideline, I mainly check:

(a) Are the claims made in the submission supported by accurate, convincing and clear evidence? and

(b) Would some individuals in TMLR's audience be interested in the findings of this paper?

**Strengths:**

- This work is generally well-written and easy to follow.
- Finding efficient optimization strategies is an important research topic in the current MTL community. This work has made a timely contribution in this direction.
- The proposed method is simple and easy to implement. It is quite promising to see it can robustly outperform other methods on both CV and robotics problems (but there are also some concerns listed below).
- The findings on task relationships obtained by Auto-\lambda are also interesting.


**Weaknesses:**

According to the strengths listed above, I have no doubt that this work could be interesting to many TMLR audiences. However, there are also some concerns about the claims and design choices made in this work.

**1. Lookahead Adaption during Training:** This work correctly mentions (Fifty et al., 2021) as a closely related work for learning the task relation, of which one limitation is its two-stage mechanism (e.g., search for the best task structure first, and then retrain the model). However, to my understanding, (Fifty et al., 2021) is one of the two ideas proposed in their previous work [1] for task relationship measuring. In [1], they also proposed a method to adjust the optimization dynamics among tasks with a lookahead strategy directly during *a single training stage*. The proposed approach in [1] can also be combined with other MTL methods.

I do see the many differences between the method proposed in [1] and Auto-\lambda, such as [1] selects gradient from a fixed set of candidates while Auto-\lambda learns the weights. But since the core design concept of Auto-\lambda is also the lookahead method, I think a more detailed discussion and comparison (similarity/difference) with the method proposed in [1] is needed.

[1] Christopher Fifty, Ehsan Amid, Zhe Zhao, Tianhe Yu, Rohan Anil, and Chelsea Finn. Measuring and harnessing transference in multi-task learning. arXiv:2010.15413, 2020.

**2. Missing Discussion on Gradient-based Hyperparameter Tuning:** If we treat the weights \lambda as hyperparameters, the proposed auto-\lambda method is closely related to the gradient-based hyperparameter adaptation approaches [2,3,4,5]. Especially, [2,5] also use gradient-based updates to adapt the hyperparameter (e.g., learning rate) at each iteration in an online manner, [6] considers the gradient-based hyperparameter adaption with the bi-level optimization formulation. What is the relation between auto-\lambda and these methods?

In addition, what if we allow different tasks to have different learning rates (might with fixed weights), and then use these methods to adapt the learning rate? Will they perform similarly with Auto-\lambda?

[2] L. B. Almeida, T. Langlois, J. D. Amaral, and A. Plakhov. Parameter adaptation in stochastic optimization. In D. Saad (ed.), On-Line Learning in Neural Networks. Cambridge University Press, 1998.

[3] Yoshua Bengio. Gradient-based optimization of hyperparameters. Neural computation 2000.

[4] Dougal Maclaurin, David Duvenaud, and Ryan Adams. Gradient-based hyperparameter optimization through reversible learning. ICML 2015.

[5] Atilim Gunes Baydin, Robert Cornish, David Martinez Rubio, Mark Schmidt, and Frank Wood. Online learning rate adaptation with hypergradient descent. ICLR 2018.

[6] Amirreza Shaban, Ching-An Cheng, Nathan Hatch, and Byron Boots. Truncated back-propagation for bilevel optimization. AISTATS 2019.

**3. Weight-based Methods v.s. Gradient-based Methods:** This work divides the multi-task optimization strategies into weight-based methods (e.g., Uncertainty/DWA/Auto-\lambda) and gradient-based methods (e.g., GradDrop, PCGrad, CAGrad). However, the current gradient-based methods (such as CAGrad and RotoGrad[7]) usually claim they consider both the scale (weight) and direction for each task's gradient. In this sense, are the weight-based method also a gradient-based method (which only considers the scale)?

Will it be more general to use the bi-level optimization and lookahead adaption to adjust the gradient (not just the weight) for each task?

[7] Adrián Javaloy, and Isabel Valera. RotoGrad: Gradient Homogenization in Multitask Learning. arXiv:2103.0263.


**4. Single Gradient Update:** In Auto-\lambda, the lookahead for model parameters \theta and weight \lambda are both updated by a single gradient update step. Will it be too myopic for a single step lookahead, and not sufficiently adjusted with only a single step update for \lambda? How about conducting multiple gradient updates for both?

**5. Comparison with Simple Method:** Some current works show that a simple MTL algorithm with random weight [8] or fixed linear scalarization with regularization [9] can robustly achieve comparable performance with the more sophisticated optimization approach (e.g., the weight-based and gradient-based methods), and they do not require any computation/memory overhead. These works also test their methods on computer vision (e.g., NYUv2, Cityscapes, PASCAL-Context, CelebA) and reinforcement learning (e.g., Meta-World) problems. It could be beneficial to clearly discuss and compare auto-\lambda with those simple approaches.

[8] Baijiong Lin, Feiyang Ye, and Yu Zhang. A Closer Look at Loss Weighting in Multi-Task Learning. arXiv:2111.10603.

[9] Vitaly Kurin, Alessandro De Palma, Ilya Kostrikov, Shimon Whiteson, and M. Pawan Kumar. In Defense of the Unitary Scalarization for Deep Multi-Task Learning. arXiv:2201.04122.

**6. More Analysis with Uniform Weighting:** According to Table 5, all weight-based/gradient-based methods (and their combination) can always outperform the vanilla approach with equal weights. However, the results in other works (e.g., [8][10]) show that many weight-based/gradient-based methods might indeed have poorer performance. Where does this inconsistency come from? Will the relative performance be very sensitive to the dataset/model/hyperparameter setting?

[10] Simon Vandenhende, Stamatios Georgoulis, Wouter Van Gansbeke, Marc Proesmans, Dengxin Dai, and Luc Van Gool. Multi-task learning for dense prediction tasks: A survey. TPAMI 2021.

**7. Weight-based/Gradient-based Method Combination:** It is quite interesting to see this combination can further improve the MTL performance (except for PCGrad + DWA). In the combination, is the gradient-based approach applied for both lookahead (eq 4) and the actual parameter update (eq 6)? Will it actually violate/change the assumption/setting for the gradient-based methods (e.g., the scale adaption step in CAGrad and RotoGrad [7]).

**8. Computational Overhead:** This work explicitly discusses its limitation on the training speed. I appreciate the clear discussion and wonder if allowing other methods to train longer (e.g., the same wall-clock time or forward/backward passes with Auto-\lambda) can close the performance gap.

---

> ### Author Response · Authors · 2022-04-27
> **Reponse for R3 (Part 2)**
>
> -- ***More Analysis with Uniform Weighting***
>
> We have followed the same data pre-processing to be consistent with prior works: MTAN [Liu, 2021], AdaShare [Sun 2020], Uncertainty [Kendall, 2018], in different datasets and trained all methods with the same hyper-parameters to make sure all comparisons are fair. We did observe that some MTL optimisation methods might have worse performance such as in CIFAR-100 -- DWA and Uncertainty perform worse than Equal. We agree that the relative performance can be sensitive to the datasets/model/hyper-parameters, and thus we have evaluated Auto-Lambda in multiple settings with different datasets/architectures to minimise this inconsistency. In addition, we have open-sourced the code and the significance of this work will be further tested by the community.
>
> -- ***Weight-based/Gradient-based Method Combination***
>
> In this original experiment, we first applied gradient-based methods and then applied weighting-based methods based on the updated gradients. And thus, it will not violate assumptions on any of these optimisation methods. This experiment is mainly inspired by the original paper of PCGrad, showing a higher performance can be achieved by coupling PCGrad with Uncertainty (compared to DWA and Equal) -- which is consistent with our findings.
>
> -- ***Trade-off of Performance with Approximation***
>
> We have included an additional experiment in the Ablative Analysis section showing that the learned weightings have nearly no difference when using first-order approximation compared to exact second-order gradients (we are only optimising very few parameters after all.) Similarly, we have observed no performance difference when having a small $K'$, but we always choose the largest possible $K'$ to fit the GPU's memory to speed-up training, and we have included this observation in Section 4.
>
>
> **Update R3:** We have added a paragraph discussing hyper-parameter optimisation in the Related Work section **[Update B]** and we have compared with some other simple weighting-based methods, with the support of training time analysis **[Update D]**. We have included another ablative analysis to compare Auto-Lambda trained with direct optimisation without approximation **[Update D].** All minor issues have been resolved.

---

> > ### Comment · Reviewer_BnS4 · 2022-05-05
> > **Follow-up Comment**
> >
> > Thank you for your thorough response, and I also appreciate the additional experiments and discussions. Most of my concerns have been appropriately addressed.
> >
> > Here are a few comments:
> >
> > **Gradient-based Hyper-parameter Tuning**
> >
> > Thank you for the clarification, and I do see the contribution of Auto-\Lambda to consider both MTL and AL via the same framework.
> >
> > It seems that learning per-task learning rates (rather than per-task weightings) in MTL can avoid reaching the trivial solutions (e.g., \lambda_i = 0). What is the advantage of meta-learning based optimization over direct first-order optimization, especially with a single-step gradient update?
> >
> > **Minor Issue**
> >
> > The citation of multiple works could be better placed in chronological order.

---

> > > ### Author Response · Authors · 2022-05-05
> > > **Response to R3's follow-up**
> > >
> > > Thanks R3 for the follow-up. And we are glad that most of your concerns have now been resolved.
> > >
> > > -- **Gradient-based Hyper-parameter Tuning**
> > >
> > > Sorry, we are not exactly following this question. Are you asking about the Auto-Lambda's formulation or meta-learning per-task learning rate's formulation? If it's for the prior case, as we have discussed that using direct first-order optimisation will lead to trivial solutions (\lambda_i=0). If it's for the latter case, we believe learning per-task learning rates will highly depend on the design of architecture design, and the choice of optimisers, rather than learning a global/general relationships as from Auto-Lambda. Learning per-task learning rate is a bit out of the scope of this work, we have no direct experience with this problem and therefore are not able to comment further.
> > >
> > > -- **Minor Issue**
> > >
> > > Thanks for the suggestion. We will fix this in the updated version of this paper.

---

> > > > ### Comment · Reviewer_BnS4 · 2022-05-07
> > > > **Regarding Per-Task Weightings v.s. Per-Task Learning Rate**
> > > >
> > > > Thank you for the further response. The question is about the latter case. My original thought was since "learning per-task learning rates is mathematically the same as learning per-task weightings", it is straightforward to rewrite the "manually defined learning rate + learnable per-task weightings" (e.g., eq (4) and (6)) into "learnable per-task learning rate + manually defined weightings (e.g., all 1)".
> > > >
> > > > The relative scale of per-task learning rates (and its dynamic nature) could play the same rule as the current per-task weightings for general task relationships. In this way, it is also possible to use direct first-order optimization on the learning rates rather than the meta-learning approach.

---

> > > > > ### Author Response · Authors · 2022-05-07
> > > > > **Response to Per-task Learning Rate**
> > > > >
> > > > > Yes, I understand your point now. It's indeed a very interesting perspective!
> > > > >
> > > > > I believe the direct first-order optimisation in such case will work in the multi-task learning setting, but it would still not work in the auxiliary learning setting, since then the gradients of the learning rates for auxiliary tasks would be zero (if we backprop from the loss of primary tasks). Eventually, it will lead to a similar degration issue but in a different way.

---

> > > > > > ### Comment · Reviewer_BnS4 · 2022-05-14
> > > > > > **Thank you**
> > > > > >
> > > > > > Thank you for the response, and I do see the unique value of Auto-\lambda for the auxiliary learning in this case. I think this flexibility could be important in practice.
> > > > > >
> > > > > > All my concerns have been properly addressed, and I lean toward accepting this work. I have made my recommendation decision accordingly.

---

> ### Author Response · Authors · 2022-04-27
> **Reponse for R3 (Part 1)**
>
> We thank R3 for carefully assessing our paper and providing constructive and detailed suggestions. And we hope to resolve some of your concerns in the following comments.
>
> -- ***Connection to Fifty 2020.***
>
> We are aware of the original work of Fifty 2020, and we only discussed its new follow-up work mainly following the suggestion from the original paper. Yes, we shared a similar motivation for using a look-ahead loss to refine MTL optimisation. However, as R3 also observed, the detailed design strategies of using look-ahead loss is different -- Fifty 2020's method was based on the computing inner task affinity score, i.e.. how one-step gradient update on shared parameters from one task can alter the performance of another task. In such design, it needs to compute each per-task affinity score for $K$ tasks) and it needs to couple with other gradient-based optimisation frameworks such as PCGrad. Whilst Auto-Lambda is a standalone optimisation framework which optimises on task-weightings solely, and with a constant compute memory. We have added some additional discussions on Fifty 2020/2021 in the updated paper (second paragraph Section 4).
>
> -- ***Connection to Hyper-parameter Optimisation***
>
> Direct first-order hyper-parameter optimisation such as in [2,3,4,5] cannot be applied in MTL problems, since we will reach trivial solutions: $\lambda_i=0$. This is one of the major reasons that most MTL optimisation frameworks using a **surrogate measurement** to compute task weightings (such as via uncertainty, or the change rate of training losses).
>
> However, Auto-Lambda is indeed related to meta-learning based hyper-parameter optimisation such as [6], since both methods formulate a bi-level optimisation problem. Learning per-task learning rates is mathematically the same as learning per-task weightings. As also mentioned in R1's comments, one of the contributions of our paper is to highlight this interesting perspective to consider multi-task learning and auxiliary learning as the same optimisation problems which can be solved via bi-level optimisation with existing/well-developed tools.
>
> We have added another paragraph discussing these works in the Related Work section.
>
> -- ***Weight-based Methods v.s. Gradient-based Methods***
>
> Though both CAGrad and RotoGrad do normalise the scale, the main purpose of their scale normalisation is part of the design to better help manipulate task gradients in a specific way. However, the weighting-based approaches like Uncertainty, DWA and Auto-Lambda just focus on learning task weightings without altering any internal gradients. So the implementation is simpler and training time is faster.
>
> It's possible to learn gradients via bi-level optimisation mathematically. But the learning space is too huge which would be not simple and timely/memory efficient enough to optimise (in Auto-Lambda, we only need to optimise K parameters for K tasks, and thus the training time is reasonable). But it is an interesting future direction.
>
> -- ***Single Gradient Update***
>
> We may approximate the number of steps as the learning rate to update $\lambda$ -- *the more steps, the higher the learning rate*. But we want to design Auto-Lambda to be *as simple as possible*, with the least number of hyper-parameters, and thus we tried to avoid introducing additional learning tricks. In practice, we found that just tuning the learning rate is sufficient to adjust the learning efficiency.
>
> -- ***Comparison with Simple Method / Computational Overhead***
>
> We agree that comparing with other baselines in a simple setting will be interesting. As such, we have added an additional experiment in [9]'s challenging setting trained with strong regularisation, and we have also compared training-time across these multi-task methods (section 7.3 in the updated paper). We found that Auto-$\lambda$ is still able to improve performance comparing to all multi-task methods evaluated in our paper and [9], and it is still an order of magnitude faster than some gradient-based methods, such as PCGrad [Yu, 2020] and CAGrad [Liu, 2021a].
>
> In addition, we have confirmed that training longer will not further improve performance -- we have tuned the hyper-parameters to make sure the performance improvement is really from the optimisation methods rather than from different hyper-parameters.

---

### Review · Reviewer_kiR7 · 2022-04-19

**Summary Of Contributions:**

The main contribution of this work is a method on learning the task combination weights dynamically in the setting of multitask learning (MTL). The proposed method could be applied in two variants of MTL. The first one is the classic setting where one hopes to obtain the optimal weighted loss; the second one is a special case of the first one, where only a subset of tasks are of main interest, whereas the others are treated as auxiliary tasks used to further improve the performance of the first one. The main strength of the proposed approach is empirical -- the authors demonstrated the effectiveness of the proposed approach on several benchmark datasets, and also did a good job in visualizing the learned task weights.


**Requested Changes:**

Please see my comments above.

**Strengths And Weaknesses:**

As I summarized above, the main strength of this work is empirical, and its applicability to the two mentioned setting of MTL. The paper is also well-written and easy to follow. However, conceptually, I have some concerns about the technical contribution of this work, and more generally, I feel there are several important claims made in the paper not well justified or clearly defined.

Since the central claim of this work is a method used to dynamically discover the relationships between different tasks, and then adjust the weights accordingly, I think it is essential to first properly define the meaning of ``task relationship''. For example, in [1] the authors defined each task to be a vector that corresponds to the optimal linear classifier of a task, and then formally define the task relationship to be the correlation scores between two different task vectors. I am not saying that the authors need to follow the exact definition, but there should be a definition about the task relationship in order to (empirically) justify the claims made in the previous sections.

From this perspective, it seems that the $\lambda$ in Eq. (3) is implicitly referred to as the task relationship. However, there are several issues with this formulation. See my comments below.
-   For the current optimization problem in Eq. (3), it is ill-defined: $\theta$ should also appear in the optimization variables.
-   Furthermore, it is unclear to me the following degenerate case could be avoided: one could make $\lambda \to 0$ and the problem is still feasible. However, when $\lambda \to 0$, the minimization problem becomes vacuous and it is unclear solving this optimization problem will lead to any meaningful model parameters.
-   In each iteration of the optimization algorithm, the $\lambda$ will be different. But if the authors implicitly treat $\lambda$ to be the task relationship, this implies that the notion of task relationship is a local one, and is subject to change according to different model parameters. However, conceptually, the task relationship should only depend on the distributions of these tasks, and should be a global concept inherent to the tasks themselves, rather than the model being used to train these tasks. This distinction is important, as it can render all the claims made in the abstract as well as the introduction section non-justifiable.

[1].    A Convex Formulation for Learning Task Relationships in Multi-Task Learning, Zhang et al.

---

> ### Author Response · Authors · 2022-04-27
> **Reponse for R2**
>
> We thank R2 for the constructive comments and we hope to clarify your understanding on task relationships and Auto-Lambda's formulation in the following response.
>
> -- ***The Definition of Task Relationships***
>
> Yes, we agree that an accurate definition of task relationships is important, which we have improved in the updated version of the paper (second paragraph Section 1). Here, learning *the task relationship* is the same as learning *which tasks should be trained together*, as consistent with task grouping literature [Standley 2020, Fifty 2021]. For
> example, we say that task A is more related to task B than task C, if the performance of task A is higher when training tasks A and B together, compared to when training tasks A and C together. And thus, exhaustively finding the optimal combination of training tasks is computationally infeasible. This is highlighted in the first and second paragraphs of the Introduction section.
>
> Different from these task grouping methods, which consider that **task relationships are static**, i.e., task should either be trained together or not trained together (task weightings are binary), we consider that **task relationships are dynamic**, which can be defined/learned via task weightings $\lambda$. Then the problem definition is to find the optimal task weightings to maximise the performance of primary tasks, and we can say the tasks with the higher weightings are more related to the primary tasks. A simple example is shown in Fig. 1.  Additionally, we have confirmed that dynamic task relationships indeed can achieve better per-task performance than using fixed task grouping, as shown in Fig 5. And we have also observed that a noise prediction task is not related to any standard tasks (near zero weighting), as shown in Fig 3., 4., which all fit human intuition.
>
> -- ***Auto-Lambda Formulation***
> >For the current optimization problem in Eq. (3), it is ill-defined: $\theta$ should also appear in the optimization variables.
>
> Sorry we cannot exactly follow your concern. $\theta$ -- the MTL network parameters -- are in the optimisation variables. It is optimised via the inner loop of this bi-level optimisation (second line of Eq. 3. with fixed $\lambda$ -- we train $\theta$ and $\lambda$ iteratively, i.e., fix one to update the other).
>
> > Furthermore, it is unclear to me the following degenerate case could be avoided: one could make $\lambda \to 0$  and the problem is still feasible. However, when  $\lambda \to 0$ , the minimization problem becomes vacuous and it is unclear solving this optimization problem will lead to any meaningful model parameters.
>
> We have highlighted that Auto-Lambda can avoid $\lambda$ degeneration by design, in the first paragraph under Eq. 7.
>
> The degenerated $\lambda$ will appear only if we optimise $\theta, \lambda$ in a **single-level optimisation**, i.e.; $\min_{\theta, \lambda}\sum_i \lambda_i\cdot L_i(f(x_i, \theta), y_i)$. In such formulation, zero loss can be achieved if we have $\lambda_i=0$. However, we can avoid this in our current bi-level optimisation design, to update $\lambda_i$ based on the **performance of primary tasks after one update on $\lambda$-weighted loss (inner loop)**. This is one of main contributions of our work, to design a framework such can **directly learn task weightings via gradients based on the validation loss of the primary tasks** (The Design Philosophy paragraph in Section 4).
>
> >In each iteration of the optimization algorithm, the $\lambda$ will be different. But if the authors implicitly treat $\lambda$ to be the task relationship, this implies that the notion of task relationship is a local one, and is subject to change according to different model parameters. However, conceptually, the task relationship should only depend on the distributions of these tasks, and should be a global concept inherent to the tasks themselves, rather than the model being used to train these tasks. This distinction is important, as it can render all the claims made in the abstract as well as the introduction section non-justifiable.
>
> The reasoning here is absolutely correct. It is exactly because $\lambda$ will be different based on different training stage, and therefore we consider Auto-Lambda is learning **dynamic task relationships**. Intuitively, this learned relationship becomes local and subject to different architecture design. However, an interesting observation is shown in Fig. 3 -- the learned weightings are nearly identical across different architectures, showing here **the learned task relationships are global**, further verifying that Auto-Lambda is indeed learning general and global task relationships which fit our assumption.
>
> **Update R2**: We have improved the definition of task relationships as well as the design intuition of Auto-Lambda formulation **[Update A]**.

---

### Review · Reviewer_P8PB · 2022-04-19

**Summary Of Contributions:**

This paper proposes a technique to automatically find the optimal weights to dynamically balance between the task-specific losses during multi-task learning. Specifically, the authors propose a bi-level optimization framework that aims to find the weight lambda for each task to minimize the validation loss of the primary task. Moreover, to avoid inefficiency resulting from the computation of second-order gradients, the authors further propose a finite different approximation method to approximate them. The proposed framework is validated on multiple benchmark datasets for dense prediction, multi-domain classification, and robot manipulation performance, and is shown to largely outperform baselines that adjust the task-specific weights using uncertainty or changes the weighting based on the relative scale of the loss.

**Requested Changes:**

- Please provide comparison against [Lee et al. 16] and [Lin et al. 19].
- Please describe the baselines more in detail.
- Please apply the proposed loss weighting scheme to state-of-the-art MTL frameworks, that have specially designed architectures for sharing.

**Strengths And Weaknesses:**

Pros
- Learning "dynamic task weighting" using bi-level optimization seems new, although I did not extensively check for existing works that aim to achieve the same goal.
- The proposed technique seems like a practical method to automatically find the weighting across the tasks in a multi-task learning scenario, given its strong performance over existing weighting methods.
- The additional analyses of the relative scale of the weights over the course of training, and the task relationships are helpful in understanding how the method works.

Cons
- The proposed method could be seen as a simple hyperparameter optimization method based on bi-level optimization. In fact, the proposed approach is a technique or trivially known to most researchers and practitioners working on multi-task learning. Thus, the paper does not provide any new insights for researchers working on multi-task learning.
- Weighting task-specific loss is only one of many problems of multi-task learning, and what affects the performance of a multi-task learner the more, are other factors such as how to share the knowledge across tasks (for deep networks, how to construct the architecture).
- The proposed method is validated against only a few existing works, missing comparison against relevant works such as [Lee et al. 16] and [Lin et al. 19].

[Lee et al. 16] Asymmetric Multi-task Learning Based on Task Relatedness and Loss, ICML 2016
[Lin et al. 19] Pareto Multi-Task Learning, NeurIPS 2019

---

> ### Author Response · Authors · 2022-04-27
> **Reponse for R1**
>
> We thank R1 for the suggestions to improve this work.
>
> -- ***Insights and Design of this work***
>
> First of all, we thank the reviewer for highlighting that our work is new in terms of algorithmic design and helpful to understand task relationships.
>
> We agree that our work is related to hyper-parameter optimisation literature based on the fact that we all solve a bi-level optimisation problem. However, this is based on a unique perspective of **rethinking multi-task learning and auxiliary learning as the same optimisation problem which can be solved via bi-level optimisation**. We aimed to design a simple and general framework based on that perspective to maximise generality and flexibility, such that the proposed Auto-Lambda framework can be applied to any MTL/AL problems with the choice of any neural architectures, and any combination of primary tasks.
>
> To the best of our knowledge, this perspective is new and well supported by the additional analysis in Section 6. We argue that this perspective is not trivially known to MTL researchers, since all previous MTL/task grouping frameworks [Standley 2020, Fifty 2021] are designed based on the assumption that **task relationships are static** (i.e., task should either be trained together or not trained together), and consider **MTL and AL as separate** optimisation problems [Liu 2019c, Navon 2021]. If the reviewer has examples to the contrary, please share references so that we can discuss further.
>
> -- ***Better MTL Architecture***
>
> We have discussed in the original related work, that MTL optimisation is an **orthogonal direction** to MTL architecture design. To show our method is robust to the choice of different MTL architectures, we have experimented with standard MTL architecture with hard-parameter sharing, as well as one state-of-the-art MTL architecture, MTAN [Liu 2021b] (Section 5.1 in the original paper). In addition, each dataset was also evaluated with different backbone networks (Section 5.2, 5.3 in the original paper; Section 7.3 in the updated paper).
>
> We understand that designing task-specific weighting loss is just one problem in the MTL field. We hope our work which provides a unique perspective of learning dynamic task relationships could provide useful insights to help MTL researchers design better MTL architectures.
>
> -- ***More baselines***
>
> We have compared with baselines: Equal, Uncertainty, DWA (in the main experiments), GradDrop, PCGrad, CAGrad (in the additional analysis) --- in total this is 6 different baselines, in the original paper. However, to further strengthen the paper, we have now additionally compared 4 other baselines in this paper [1]'s setting, trained with a very challenging dataset with strong regularisation (section 7.3 in the updated paper).
>
> **Update R1**: We have included additional experiments with more baselines **[Update D]**, and have described the baselines in more details, as suggested by R1 **[Update A]**.
>
> [1] Vitaly Kurin et al. Defense of the Unitary Scalarization for Deep Multi-Task Learning. arXiv:2201.04122.

---

> > ### Comment · Reviewer_P8PB · 2022-05-05
> > **Not all previous MTL/task grouping frameworks are designed based on the static task relationship assumption.**
> >
> > **To the best of our knowledge, this perspective is new and well supported by the additional analysis in Section 6. We argue that this perspective is not trivially known to MTL researchers, All previous MTL/task grouping frameworks [Standley 2020, Fifty 2021] are designed based on the assumption that task relationships are static.**
> >
> > It is not true that all previous MTL/task grouping frameworks are designed based on the assumption that task relationships are static. It is known trivially to most MTL researchers that tasks in a multi-task learning have have unequal relationships, even before deep learning has become popular [Zhang and Yeung 10, Leen et al. 12]. [Lee et al. 16] proposed an asymmetric MTL method based on that specific motivation and proposed a method which performs asymmetric knowledge transfer across related tasks, by considering the task-specific loss for each task. It also does consider dynamically changing relationships among the tasks, since the loss will dynamically change during the training step. On the other hand, the proposed Auto-Lambda cannot model asymmetric relationships among tasks since it only learns the global weight on the task-specific losses. The follow-up work [Lee et al. 18] extends [Lee et al. 16] to deep representation learning, and [Nguyen et al. 21] uses the task-averaged uncertainty rather than loss to learn the asymmetric knowledge transfer structure, and are also highly relevant baselines, as they all model dynamically changing relationships among tasks.
> >
> >
> > [Zhang and Yeung 10] A Convex Formulation for Learning Task Relationships in Multi-task Learning, UAI 2010
> >
> > [Leen et al. 12] Focused Multi-task Learning in a Gaussian Process Framework, Machine Learning, 2012
> >
> > [Lee et al. 18] Deep Asymmetric Multi-task Feature Learning, ICML 2018.
> >
> > [Nguyen et al. 21] Clinical Risk Prediction with Temporal Probabilistic Asymmetric Multi-Task Learning, AAAI 2021.

---

> > > ### Author Response · Authors · 2022-05-05
> > > **Response to Novelty Concern (Part 2)**
> > >
> > > -- **Asymmetric Relationships**
> > >
> > > In Section 6, we have explored the relationships learned from Auto-Lambda, and concluded with three observations. The first two observations: "Task relationships are consistent" and "Task relationships are asymmetric" are well supported in prior works which we agree and have already mentioned in the original paper -- "This observation is supported/consistent with XXX". The unique observation is the third observation: "Task relationships are dynamic", which we believe is a new finding in the MTL community. We never say that the finding of the asymmetric relationship is novel, but rather we use this well-studied finding to confirm that Auto-Lambda does learn task relationships consistent with other approaches. Finally, Auto-Lambda can model asymmetric relationships by measuring the relative task weightings based on a different choice of primary tasks, as visualised in Fig. 4. Therefore, we don't exactly follow your concern here claiming "Auto-Lambda cannot model asymmetric relationships among tasks since it only learns the global weight on the task-specific losses".
> > >
> > > --**Connection to [Lee et al. 18, Nguyen et al. 21]**
> > > Both of these works are NOT proposing multi-task optimisation methods, but rather are proposing multi-task architectures **with the design motivated by the assumption of asymmetric relationships exsist in MTL**. In [Lee et al 20], the architecture design was motivated by finding reliable task-specific losses to avoid the negative transfer. In [Nguyen, et al 21], the architecture design was motivated by learning both in-task (different time steps) and cross-task asymmetric relationships of sequential data; the "uncertainty" is used to learn attention weights which are part of the network design, and has no connections to task relationships. The term "relationship" used in both papers is a general concept, which **motivates** the design of proposed architecture, rather than a clear metric defined in our work which **measures** how tasks are related to each other. Similarly to papers R1 mentioned in the first response, there were no discussion on relationships in both of these works, nor were evaluated in the auxiliary learning setting. Therefore, again, we don't think these works are related to Auto-Lambda.

---

> > ### Comment · Reviewer_P8PB · 2022-05-05
> > **Regarding novelty**
> >
> > **However, this is based on a unique perspective of rethinking multi-task learning and auxiliary learning as the same optimisation problem which can be solved via bi-level optimisation.**
> >
> > I do not agree that this is a unique perspective. [Alesiani et al. 20] proposed to use bilevel optimization to learn the relationships among the tasks in a multi-task learning framework as a graph, and [Lee et al. 20] used bilevel optimization framework to meta-learn the weight for each task and class-specific gradients, which basically has the same effect as learning the global weight for the task-specific loss. [Killamsetty et al. 22] propose a nested bilevel optimization framework to learn the weight of each task in meta-learning.
> >
> >
> > [Alesiani et al. 20] Towards Interpretable Multi-Task Learning Using Bilevel Programming, ECML PKDD 2020
> >
> > [Lee et al. 20] Learning to Balance: Bayesian Meta-Learning for Imbalanced and Out-of-distribution Tasks, ICLR 2020
> >
> > [Killamsetty et al. 22] A Nested Bi-level Optimization Framework for Robust Few Shot Learning, AAAI 2022.

---

> > > ### Author Response · Authors · 2022-05-05
> > > **Response to Novelty Concern**
> > >
> > > Thanks R1 for the follow-up and for highlighting these related works.
> > >
> > > However, we respectively disagree with your arguments, explained below.
> > >
> > > First of all, these mentioned related works were evaluated in a *completely different MTL setup* from our work, which is not comparable. In [Alesiani et al 20], the relationship is defined by the *structure of a graph*, and each graph node is a task. It is different from our work which defines the relationship of tasks as *what extent of these tasks should be trained together*. In [Lee et al 20, Kilamsetty et al 22], they were both evaluated in a **few-shot learning** setting, from which they were learning general representations as a form of multi-task supervised pre-training. As such, both of these works assume the test tasks are not in the training distribution, which is different from our assumption. We have already discussed this type of work in the Updated Related Work section (Meta Learning for MTL Paragraph), as similar to what R5 suggested.
> > >
> > > Secondly, as highlighted in your comment, we consider our unique perspective is to **rethink multi-task learning and auxiliary learning as the same optimisation problem**. None of the three papers you mentioned discussed/explored task relationships (Alesiani 20 did mention task relationships, but as highlighted in the previous paragraph, their task relationship is not the same definition as our task relationship), nor were evaluated in an auxiliary learning setting. Therefore, we don't consider that these three works can bring up Auto-Lambda's novelty concern.

---

### Comment · Action_Editors · 2022-04-24
**Rebuttal**

Dear authors,

We have collected five reviews. There are some concerns raised. Can you try to address the concerns by providing rebuttals as soon as possible? Thank you.

Best wishes,
Tongliang

---

### Author Response · Authors · 2022-04-27
**General Comments**

We thank all of the reviewers for their time and effort in providing these helpful comments. For simplicity, we rename Reviewers P8PB, kiR7, BnS4, QXX5, EZCk to R1, R2, R3, R4, R5 respectively. Based on these reviews, we have revised our paper and uploaded the new version. New text in the paper is written in a **purple colour**. For each reviewer, we provide a detailed, individual response to their review, including our specific changes to the paper based on their review.

To summarise, we have made the following changes to the paper:

**Update A: Improved Paper Details** Introduce baseline methods in more details (R1);  Formally define task relationships (R2, R5); Improve Auto-Lambda formulation (R2, R4, R5).

**Update B: Improved Related Work** Discussion on hyper-parameter optimisation (R1, R3, R5); Discussion on Meta-Learning based Task scheduler (R5); Discussion on Fifty 2020/2021 (R3).

**Update C: Code Release** We have already included the code in the supplementary material, and will open-source on GitHub once accepted  (R3, R4).

**Update D: Additional Experiments** More baselines in a simple setting with training time analysis (R1, R3); Different training tricks of Auto-Lambda: direct vs approximate gradients (R3).

---

### Decision · Action_Editors · 2022-05-10

**Recommendation:** Accept with minor revision

**Comment:**

The paper proposes to automatically and dynamically find optimal weights for tasks in the multiple task learning setting. A bi-level optimization framework has been used to update the weights. The proposed method has been empirically verified on several datasets and applications. Outstanding empirical performance is the main strength of the proposed method.
Overall, the claims are supported by clear empirical evidence. The good empirical results will interest many practitioners and some researchers in the community.

However, there are concerns raised regarding the soundness of claims. Several important claims made in the paper are weak or do not stand. For example, the proposed gradient-based hyperparameter tuning of task-specific losses in a multi-task learning objective is not new and very similar approaches have been exploited in existing works on MTL such as [Alesiani et al. 20], and meta-learning [Lee et al. 20, Kilamsetty et al. 22], and the related work mentioned by Reviewer BnS4. The authors claim that all existing MTL methods consider the task relationships as static, which is not true. [Lee et al. 16, Lee et al. 18, Nguyen et al. 21] also learn knowledge transfer weights in a dynamic manner. Reviewer KiR7 points out that the definition of the task relationship should be more rigorously discussed. The authors’ reply that the learned task relationships are global based on an observation of Fig. 3 is not acceptable or convincing.

We think the concerns would be addressed via a minor revision and would provide an opportunity for the authors to carefully revise the paper.

Reference:
[Alesiani et al. 20] Towards Interpretable Multi-Task Learning Using Bilevel Programming, ECML PKDD 2020 [Lee et al. 20] Learning to Balance: Bayesian Meta-Learning for Imbalanced and Out-of-distribution Tasks, ICLR 2020 [Killamsetty et al. 22] A Nested Bi-level Optimization Framework for Robust Few Shot Learning, AAAI 2022 [Lee et al. 16] Asymmetric Multi-task Learning Based on Task Relatedness and Loss, ICML 2016 [Lee et al. 18] Deep Asymmetric Multi-task Feature Learning, ICML 2018 [Nguyen et al. 21] Clinical Risk Prediction with Temporal Probabilistic Asymmetric Multi-Task Learning, AAAI 2021.

---

> ### Author Response · Authors · 2022-05-30
> **Camera-ready Version**
>
> Dear Action Editors,
>
> We have uploaded our de-anonymous camera-ready version of the paper. And we have included all of the following changes to this final version:
>
> 1. A clear definition of task relationship.
> 2. Improved Related Work: -- Meta Scheduler for Multi-task Supervised Pre-training; -- Meta-Learning for Hyper-parameter Tuning.
> 3. Added Lee et al 16/18 for related asymmetric MTL.
> 4. Training speed analysis.
> 5. Additional experiments with strong regularisation.
>
> Hope this resolves all the concerns from the reviewers.